# Out-of-Distribution Learning with Human Feedback

**Haoyue Bai**  *baihaoyue@cs.wisc.edu*
*Department of Computer Sciences*
*University of Wisconsin-Madison*

**Xuefeng Du**  *xfdu@cs.wisc.edu*
*Department of Computer Sciences*
*University of Wisconsin-Madison*

**Katie Rainey**  *krainey@niwc.navy.mil*
*Naval Information Warfare Center Pacific*

**Shibin Parameswaran**  *paramesw@spawar.navy.mil*
*Naval Information Warfare Center Pacific*

**Yixuan Li**  *sharonli@cs.wisc.edu*
*Department of Computer Sciences*
*University of Wisconsin-Madison*

**Reviewed on OpenReview:** *https://openreview.net/forum?id=5qo8MF3QU1*

## Abstract

Out-of-distribution (OOD) learning often relies on strong statistical assumptions or prede-fined OOD data distributions, limiting its effectiveness in real-world deployment for both OOD generalization and detection, especially when human inspection is minimal. This paper introduces a novel framework for OOD learning that integrates human feedback to enhance model adaptation and reliability. Our approach leverages freely available unlabeled data in the wild, which naturally captures environmental test-time OOD distributions under both covariate and semantic shifts. To effectively utilize such data, we propose selectively acquiring human feedback to label a small subset of informative samples. These labeled samples are then used to train both a multi-class classifier and an OOD detector. By incorporating human feedback, our method significantly improves model robustness and precision in handling OOD scenarios. We provide theoretical insights by establishing generalization error bounds for our algorithm. Extensive experiments demonstrate that our approach out-performs state-of-the-art methods by a significant margin. Code is publicly available at `https://github.com/HaoyueBaiZJU/ood-hf`.

## 1 Introduction

Modern machine learning models deployed in the wild can inevitably encounter shifts in data distributions. In practice, data shifts can often exhibit heterogeneous forms. For example, out-of-distribution (OOD) data can arise either from semantic shifts where the test data comes from novel categories (Yang et al., 2021), or covariate shifts where the data undergoes domain or environmental changes (Chapaneri & Jayaswal, 2022; Zhou et al., 2022a; Koh et al., 2021; Ye et al., 2022). The nature of mixed types of OOD data poses significant challenges in both OOD generalization and OOD detection—necessitating correctly predicting the covariate-shifted OOD samples into one of the known classes, while rejecting the semantic OOD data. For a model to be considered robust and reliable, it must excel in OOD generalization and OOD detection simultaneously, to ensure the continued success and safety of machine learning applications in real-world environments.

Previous works on OOD learning often rely heavily on statistical approaches (Ye et al., 2023; Haroush et al., 2021) or predefined assumptions (Ye et al., 2021; Zhang et al., 2024) about OOD data distributions, which may not accurately reflect the complexity and diversity of real-world scenarios. Consequently, they struggle to adapt to unforeseen OOD distributions encountered during deployment effectively. Furthermore, without human input to provide contextual information and guide model adaptation, these approaches may face challenges in accurately distinguishing between in-distribution (ID) and OOD data, leading to suboptimal performance in OOD detection tasks. The absence of human feedback thus restricts the adaptability of previous approaches, hindering their efficacy in addressing the multifaceted challenges of OOD generalization and detection in real-world deployment environments.

To tackle the challenge, we introduce a novel framework for OOD learning with human feedback, which can provide valuable insights into the nature of OOD shifts and guide effective model adaptation. Human feedback offers a unique perspective that complements automated statistical techniques, and remains under-explored in the context of OOD learning. Our framework capitalizes on the abundance of unlabeled data available in the wild, capturing environmental OOD distributions under diverse conditions, which can be characterized as a composite mixture of ID, covariate OOD, and semantic OOD data (Bai et al., 2023). Such unlabeled data is ubiquitous in many real-world applications, arising organically and freely in the model's operational environment. To harness the unlabeled data for OOD learning, our key idea is to selectively provide human feedback and label a small number of highly informative samples from the wild data distribution, which are then used to train a robust multi-class classifier that generalizes across different covariate OOD samples and a reliable OOD detector that can identify the semantic OOD data points. By exploiting human feedback, we can enhance the robustness and reliability of machine learning models, equipping them with the capability to handle OOD scenarios with greater precision.

Our framework employs a gradient-based sample selection mechanism, which prioritizes the most informative samples for human feedback (Section 3.1). The sampling score is calculated based on the projection of its gradient onto the top singular vector of the gradient matrix, defined over all the unlabeled wild data. Specifically, the sampling score measures the norm of the projected vector, which can be used to select informative samples (e.g., ones with relatively large gradient-based scores). The selected samples are then annotated by human, and incorporated into our learning framework. In training, we jointly optimize for both robust classification of samples from the ID and the annotated covariate OOD data, along with a reliable binary OOD detector separating between the ID data and the annotated semantic OOD data (Section 3.2). Additionally, we deliver theoretical insights (Theorem 1) into the learnability of the classifier with the gradient-based sampling score, thus formally justifying the framework of OOD learning with human feedback.

Lastly, we provide extensive experiments showing that this human-centered approach can effectively improve both OOD generalization and detection under a small annotation budget (Section 4). Compared to SCONE (Bai et al., 2023), the current state-of-the-art method, we substantially improve the accuracy of OOD classification by 5. 82% on covariate-shifted CIFAR-10 data, while reducing the average error of OOD detection by 15.16% (FPR95). Moreover, we provide comprehensive ablations on the impacts of labeling budgets, different sampling scores, and sampling strategies, which leads to an improved understanding of OOD learning with human feedback.

Our framework extends active learning literature (Li et al., 2024) which focuses on ID classification to joint OOD generalization and detection tasks by leveraging wild unlabeled data. This challenging and heterogeneous mixture of data points requires an effective and tailored gradient-based strategy for sample selection and training, which improves over existing active learning strategies, as shown in our experiments (Section 4.3). To summarize our key contributions:

- We propose a new OOD learning framework with human feedbacks for joint OOD generalization and detection. Our method employs a gradient-based sampling procedure, which can select informative semantic and covariate OOD data from the wild data for OOD learning.
- We present extensive empirical analysis and ablation studies to understand our learning framework. The results provide insights into using human feedback on the unlabeled wild data for both OOD generalization and detection and justify the efficacy of our algorithm.
- We provide a generalization error bound for the model learned under human feedback, theoretically supporting our proposed algorithm.

## 2 Problem Setup

**Labeled in-distribution data.** Let $\mathcal{X}$ denote the input space and $\mathcal{Y} = \{1, ..., C\}$ denote the label space for ID data. We use $\mathbb{P}_{\mathcal{X}\mathcal{Y}}$ as the ID joint distribution defined over $\mathcal{X} \times \mathcal{Y}$. Given an ID joint distribution $\mathbb{P}_{\mathcal{X}\mathcal{Y}}$, the labeled ID data $\mathcal{S}^{\text{in}} = \{(\mathbf{x}_i, y_i)\}_{i=1}^n$ is drawn independently and identically (i.i.d.) from $\mathbb{P}_{\mathcal{X}\mathcal{Y}}$.

**Unlabeled wild data.** Upon deploying a classifier trained on ID, we have access to unlabeled data from the wild, denoted as $\mathcal{S}_{\text{wild}} = \{\tilde{\mathbf{x}}_i\}_{i=1}^m$, which can be used to assist in OOD learning. Following (Bai et al., 2023), $\mathcal{S}_{\text{wild}}$ is drawn i.i.d. from an unknown wild distribution $\mathbb{P}_{\text{wild}}$ defined below.

**Definition 1.** *The marginal distribution of the wild data is defined as:*

$$\mathbb{P}_{wild} := (1 - \pi_c - \pi_s)\mathbb{P}_{in} + \pi_c \mathbb{P}_{out}^{covariate} + \pi_s \mathbb{P}_{out}^{semantic},$$

*where $\pi_c, \pi_s, \pi_c + \pi_s \in [0, 1]$. $\mathbb{P}_{in}$, $\mathbb{P}_{out}^{covariate}$, and $\mathbb{P}_{out}^{semantic}$ represent the marginal distributions of ID, covariate-shifted OOD, and semantic-shifted OOD data respectively.*

**Learning goal.** Our learning framework aims to build a robust multi-class predictor $f_{\mathbf{w}}$ and an OOD detector $D_{\boldsymbol{\theta}}$ by leveraging knowledge from labeled ID data $\mathcal{S}^{\text{in}}$ and unlabeled wild data $\mathcal{S}_{\text{wild}}$. Moreover, we allow a maximum number of $k$ human annotations for samples in the unlabeled data. Let $f_{\mathbf{w}} : \mathcal{X} \mapsto \mathbb{R}^C$ be a multi-class predictor with parameter $\mathbf{w} \in \mathcal{W}$, where $\mathcal{W}$ is the parameter space. The predicted label for an input $\mathbf{x}$ is

$$\widehat{y}(\mathbf{x}; f_{\mathbf{w}}) := \underset{y \in \mathcal{Y}}{\arg\max}\, f_{\mathbf{w}}^y(\mathbf{x}),$$

where $f_{\mathbf{w}}^y$ is the $y$-th coordinate of $f_{\mathbf{w}}$, and $\mathbf{x}$ can be either ID or covariate OOD. To detect the semantic OOD data, we need to construct a ranking function $g_{\boldsymbol{\theta}} : \mathcal{X} \to \mathbb{R}$ with parameter $\boldsymbol{\theta} \in \boldsymbol{\Theta}$, where $\boldsymbol{\Theta}$ is the parameter space. With the ranking function $g_{\boldsymbol{\theta}}$, one can define the OOD detector:

$$D_{\boldsymbol{\theta}}(\mathbf{x}; \lambda) := \begin{cases} \text{ID} & \text{if } g_{\boldsymbol{\theta}}(\mathbf{x}) > \lambda, \\ \text{OOD} & \text{if } g_{\boldsymbol{\theta}}(\mathbf{x}) \le \lambda, \end{cases} \tag{1}$$

where $\lambda$ is a threshold, typically chosen so that a high fraction of ID data is correctly classified.

## 3 Proposed Framework

In this section, we introduce a novel framework for OOD learning with human feedback for tackling both OOD generalization and OOD detection problems jointly. Our framework is motivated by the fundamental challenge in harnessing unlabeled wild data for OOD learning—the lack of supervision for samples drawn from the wild data distribution $\mathbb{P}_{\text{wild}}$. To address this challenge, our key idea is to selectively label a small number of samples from the wild data distribution to train a robust multi-class classifier and an OOD detector. Specifically, the design of our framework constitutes two components revolving around the following unexplored questions:

**Q1:** *How to select informative samples from the unlabeled data for human feedback?* (Section 3.1)

**Q2:** *How to learn from these newly labeled samples to enhance OOD generalization and OOD detection capabilities?* (Section 3.2)

### 3.1 Sample Selection for Human Feedback

The key to our OOD learning with human feedback framework lies in a sample selection procedure that identifies the most informative samples while reducing labeling costs. With a limited labeling budget, it is advantageous to select samples from wild data that are either covariate OOD or semantic OOD and will contribute the most to OOD generalization and detection purposes. These samples would be informative for the purpose of OOD generalization and OOD detection. Given a heterogeneous set of wild unlabeled data $\mathcal{S}_{\text{wild}}$, our rationale is to employ a sampling score that can effectively separate ID vs. non-ID part. This way, we can accordingly query samples from the non-ID pool that are most likely covariate or semantic OOD. To achieve this, we proceed to describe the sampling score.

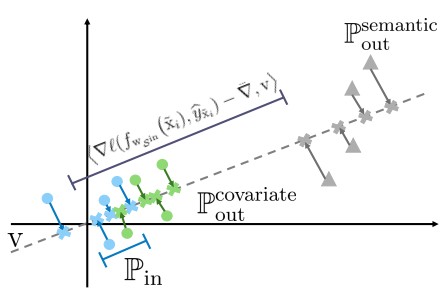 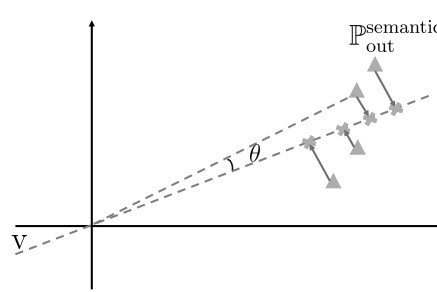

(a) Visualization of the gradient vectors and their projection

(b) Angle between the gradient and the singular vector

Figure 1: Illustration of the gradient vectors and their projections (the blue points denote $\mathbb{P}_{\text{in}}$, the green points represent $\mathbb{P}_{\text{out}}^{\text{covariate}}$, and the gray points indicate $\mathbb{P}_{\text{out}}^{\text{semantic}}$): (a) Visualization of the gradient projected onto the top singular vector of matrix $\mathbf{G}$ for unlabeled data. The gradients of the set $\mathbb{P}_{\text{in}}$ (inliers in the wild) are proximate to the origin (reference gradient $\bar{\nabla}$), in contrast to the gradients of the set $\mathbb{P}_{\text{out}}^{\text{semantic}}$, which are more distant. (b) The angle $\theta$ between the gradient of the set $\mathbb{P}_{\text{out}}^{\text{semantic}}$ and the singular vector $\mathbf{v}$. As $\mathbf{v}$ is identified to maximize the distance of the projected points (denoted by ✖) from the origin, considering the sum over all the gradients in $\mathbb{P}_{\text{wild}}$, $\mathbf{v}$ indicates the direction of OOD data in the wild with a small angle $\theta$.

**Sampling score.** We employ a gradient-based sampling score, where the gradients are estimated from a classification predictor $f_{\mathbf{w}_{\mathcal{S}^{\text{in}}}}$ trained on the ID data $\mathcal{S}^{\text{in}}$:

$$\mathbf{w}_{\mathcal{S}^{\text{in}}} \in \arg\min_{\mathbf{w}\in\mathcal{W}} R_{\mathcal{S}^{\text{in}}}(f_{\mathbf{w}}), \tag{2}$$

where $R_{\mathcal{S}^{\text{in}}}(f_{\mathbf{w}}) = \frac{1}{n}\sum_{(\mathbf{x}_i,y_i)\in\mathcal{S}^{\text{in}}} \ell(f_{\mathbf{w}}(\mathbf{x}_i), y_i)$, $\ell : \mathbb{R}^C \times \mathcal{Y} \to \mathbb{R}^+$ is the loss function, $\mathbf{w}_{\mathcal{S}^{\text{in}}}$ is the learned parameter and $n$ is the size of ID training set. The average gradient $\bar{\nabla}$ is:

$$\bar{\nabla} = \frac{1}{n} \sum_{(\mathbf{x}_i,y_i)\in\mathcal{S}^{\text{in}}} \nabla\ell(f_{\mathbf{w}_{\mathcal{S}^{\text{in}}}}(\mathbf{x}_i), y_i), \tag{3}$$

where $\bar{\nabla}$ acts as a reference gradient that allows measuring the deviation of any other points from it.

With the reference gradient defined, we can now represent each point in $\mathcal{S}_{\text{wild}}$ as a gradient vector, relative to the reference gradient $\bar{\nabla}$. Specifically, we calculate the gradient matrix (after subtracting the reference gradient $\bar{\nabla}$) for the wild data as follows:

$$\mathbf{G} = \left[ \begin{array}{c} \nabla\ell(f_{\mathbf{w}_{\mathcal{S}^{\text{in}}}}(\tilde{\mathbf{x}}_1), \widehat{y}_{\tilde{\mathbf{x}}_1}) - \bar{\nabla} \\ ... \\ \nabla\ell(f_{\mathbf{w}_{\mathcal{S}^{\text{in}}}}(\tilde{\mathbf{x}}_m), \widehat{y}_{\tilde{\mathbf{x}}_m}) - \bar{\nabla} \end{array} \right]^{\top}, \tag{4}$$

where $m$ denotes the size of the wild dataset, and $\widehat{y}_{\tilde{\mathbf{x}}}$ is the predicted label for a wild sample $\tilde{\mathbf{x}}$[1].

For each data point $\tilde{\mathbf{x}}_i$ in $\mathcal{S}_{\text{wild}}$, we now define our gradient-based sampling score as follows:

$$\tau_i = \left\langle \nabla\ell(f_{\mathbf{w}_{\mathcal{S}^{\text{in}}}}(\tilde{\mathbf{x}}_i), \widehat{y}_{\tilde{\mathbf{x}}_i}) - \bar{\nabla}, \mathbf{v} \right\rangle^2, \tag{5}$$

where $\langle\cdot,\cdot\rangle$ is the dot product operator and $\mathbf{v}$ is the top singular vector of $\mathbf{G}$. The top singular vector $\mathbf{v}$ can be regarded as the principal component of the matrix $\mathbf{G}$ in Eq. 4, which maximizes the total distance from the projected gradients (onto the direction of $\mathbf{v}$) to the origin (sum over all points in $\mathcal{S}_{\text{wild}}$) (Hotelling, 1933). Specifically, $\mathbf{v}$ is a unit-norm vector and can be computed as follows:

$$\mathbf{v} \in \arg\max_{\|\mathbf{u}\|_2=1} \sum_{\tilde{\mathbf{x}}_i\in\mathcal{S}_{\text{wild}}} \left\langle \mathbf{u}, \nabla\ell(f_{\mathbf{w}_{\mathcal{S}^{\text{in}}}}(\tilde{\mathbf{x}}_i), \widehat{y}_{\tilde{\mathbf{x}}_i}) - \bar{\nabla} \right\rangle^2. \tag{6}$$

Essentially, the sampling score $\tau_i$ in Eq. 5 measures the $\ell_2$ norm of the projected vector. To help readers better understand our design rationale, we provide an illustrative example of the gradient vectors and their projections in Figure 1 (see caption for details).

---

[1]The shape of matrix $\mathbf{G}$ is $\dim(\mathbf{w}_{\mathcal{S}^{\text{in}}}) \times m$ where each entry of $\mathbf{G}$ has a dimension of $\dim(\mathbf{w}_{\mathcal{S}^{\text{in}}}) \times 1$. In practice, $\dim(\mathbf{w}_{\mathcal{S}^{\text{in}}})$ is equal to the dimension of the penultimate layer embeddings.

**Sampling strategy.** Given the gradient-based scores calculated for each sample $\tilde{\mathbf{x}}_i$ in $\mathcal{S}_{\text{wild}}$, we need to select a subset of $k \ll m$ examples for manual labeling. Here $k$ is the annotation budget. We consider three sampling strategies, as illustrated in Figure 2.

- **Top-$k$ sampling**: select $k$ samples from $\mathcal{S}_{\text{wild}}$ with the largest score $\tau_i$. As shown in Figure 2 (a), these samples deviate mostly from the ID data and are more obviously to be semantic OOD or covariate OOD.

- **Near-boundary sampling**: select $k$ samples that are close to the ID boundary, which may encompass samples with high ambiguity. As shown in Figure 2 (b), we choose the threshold $\tau_b$ using labeled ID data $\mathcal{S}^{\text{in}}$ so that it captures a substantial fraction (e.g., 95%) of ID samples. Based on the threshold $\tau_b$, we then select $k$ samples from $\mathcal{S}_{\text{wild}}$ that are closest to this threshold.

- **Mixed sampling**: select samples using both top-$k$ and near-boundary sampling, and combine the two subsets.

Without loss of generality, we denote the selected set of samples as $\mathcal{S}_{\text{selected}}$, with cardinality $|\mathcal{S}_{\text{selected}}| = k$. For each sample in $\mathcal{S}_{\text{selected}}$, we ask the human annotator to choose a label from $\mathcal{Y} \cup \{\bot\}$, where $\{\bot\}$ indicates the semantic OOD. For covariate OOD, the returned label belongs to the existing label space $\mathcal{Y}$ according to the definition. In Section 4.3, we perform comprehensive ablations to understand the efficacy of each sampling strategy.

## 3.2 Learning Objective Leveraging Human Feedback

We now discuss our learning objective, which incorporates the human annotated samples from wild data. For notation convenience, we use $\mathcal{S}^c_{\text{selected}}$ and $\mathcal{S}^s_{\text{selected}}$ to denote labeled samples corresponding to covariate OOD and semantic OOD respectively, where $\mathcal{S}_{\text{selected}} = \mathcal{S}^c_{\text{selected}} \cup \mathcal{S}^s_{\text{selected}}$. Our learning framework jointly optimizes for both: (1) robust classification of samples from $\mathcal{S}^{\text{in}}$ and covariate OOD $\mathcal{S}^c_{\text{selected}}$, and (2) reliable binary OOD detector separating data between $\mathcal{S}^{\text{in}}$ and semantic OOD $\mathcal{S}^s_{\text{selected}}$. Given a weighting factor $\alpha$, the risk can be formalized as follows:

$$\mathbf{w}, \boldsymbol{\theta} = \arg\min[\underbrace{R_{\mathcal{S}^{\text{in}}, \mathcal{S}^c_{\text{selected}}}(f_{\mathbf{w}})}_{\text{Multi-class classifier}} + \alpha \cdot \underbrace{R_{\mathcal{S}^{\text{in}}, \mathcal{S}^s_{\text{selected}}}(g_{\boldsymbol{\theta}})}_{\text{OOD detector}}], \tag{7}$$

where the first term can be empirically optimized using the standard cross-entropy loss. The second term can be viewed as explicitly optimizing the level-set based on the model output (threshold at 0), where the labeled ID data $\mathbf{x}$ from $\mathcal{S}^{\text{in}}$ has positive values and vice versa:

$$\begin{aligned} R_{\mathcal{S}^{\text{in}}, \mathcal{S}^s_{\text{selected}}}(g_{\boldsymbol{\theta}}) &= R^+_{\mathcal{S}^{\text{in}}}(g_{\boldsymbol{\theta}}) + R^-_{\mathcal{S}^s_{\text{selected}}}(g_{\boldsymbol{\theta}}) \\ &= \mathbb{E}_{\mathbf{x} \in \mathcal{S}^{\text{in}}} \ \mathbb{1}\{g_{\boldsymbol{\theta}}(\mathbf{x}) \leq 0\} \\ &\quad + \mathbb{E}_{\tilde{\mathbf{x}} \in \mathcal{S}^s_{\text{selected}}} \ \mathbb{1}\{g_{\boldsymbol{\theta}}(\tilde{\mathbf{x}}) > 0\}. \end{aligned} \tag{8}$$

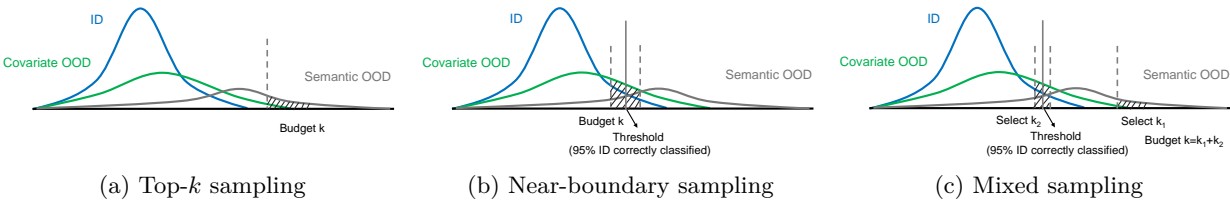

(a) Top-$k$ sampling    (b) Near-boundary sampling    (c) Mixed sampling

Figure 2: Illustration of three selection criteria, (1) top-$k$ sampling, (2) near-boundary sampling, and (3) mixed sampling. The horizontal axis is the sampling score defined in Equation 5, and the vertical axis is the frequency. Note that we color the three different sub-distributions (ID, covariate OOD, semantic OOD) separately for clarity, but in practice, the membership is not revealed due to the unlabeled nature of wild data.

To make the 0/1 loss tractable, we replace it with the binary sigmoid loss, a smooth approximation of the 0/1 loss. We train $g_{\boldsymbol{\theta}}$ along with the multi-class classifier $f_{\mathbf{w}}$. The training enables generalization to OOD samples drawn from $\mathbb{P}_{\text{out}}^{\text{covariate}}$, and at the same time, teaches the OOD detector to identify data from $\mathbb{P}_{\text{out}}^{\text{semantic}}$. The above process of sample selection, human annotation, and model training can be repeated until the desired performance level is achieved or the entire budget allocated for annotations is exhausted. An end-to-end algorithm is fully specified in Appendix A.

**Theoretical insights.** We now present theory to support our proposed algorithm. Our main Theorem 1 provides a generalization error bound *w.r.t.* the empirical multi-class classifier $f_{\mathbf{w}}$, learned on ID data and the selected covariate OOD data by the objective $R_{\mathcal{S}^{\text{in}}, \mathcal{S}^{\text{c}}_{\text{selected}}}(f_{\mathbf{w}})$. We specify several mild assumptions and necessary notations for our theorem in Appendix I. Due to space limitations, we omit unimportant constants and simplify the statements of our theorems. We defer the full formal statements to Appendix J. All proofs can be found in Appendices K and L.

**Theorem 1.** *(Informal). Let $\mathcal{W}$ be a hypothesis space with a VC-dimension of $d$. Denote the datasets $\mathcal{S}^{in}$ and $\mathcal{S}^{c}_{selected}$ as the labeled ID and the selected covariate OOD data by our learning algorithm, and their sizes are $n$ and $m_c$, respectively. If $\widehat{\mathbf{w}} \in \mathcal{W}$ minimizes the empirical risk $R_{\mathcal{S}^{in}, \mathcal{S}^{c}_{selected}}(f_{\mathbf{w}})$ for classifying the ID and covariate OOD data, and $\mathbf{w}^{*} = \arg\min_{\mathbf{w} \in \mathcal{W}} R_{\mathbb{P}^{covariate}_{out}}(f_{\mathbf{w}})$, then for any $\delta \in (0,1)$, with probability of at least $1 - \delta$, we have*

$$R_{\mathbb{P}^{covariate}_{out}}(f_{\widehat{\mathbf{w}}}) \leq R_{\mathbb{P}^{covariate}_{out}}(f_{\mathbf{w}^{*}}) + 2 \sup_{\mathbf{w} \in \mathcal{W}} d^{\ell}_{\mathbf{w}}(\mathcal{S}^{in}, \mathcal{S}^{c}_{selected})$$

$$+ 4\sqrt{\frac{2d \log(2m_c) + \log \frac{2}{\delta}}{m_c}} + 2\gamma + 2\zeta,$$

*where $\zeta = \sqrt{(\frac{1}{n} + \frac{1}{m_c})(\frac{d \log (2n + 2m_c) - \log(\delta)}{2})} + M$ and $\gamma = \min_{\mathbf{w} \in \mathcal{W}} R_{\mathbb{P}_{in}}(f_{\mathbf{w}})$. $M$ is the upper bound of the loss function for the multi-class classifier,*

$$d^{\ell}_{\mathbf{w}}(\mathcal{S}^{in}, \mathcal{S}^{c}_{selected}) = \|\nabla R_{\mathcal{S}^{in}}(f_{\mathbf{w}}, \widehat{f}) - \nabla R_{\mathcal{S}^{c}_{selected}}(f_{\mathbf{w}}, \widehat{f})\|_2,$$

*where $\widehat{f}$ is a classifier which returns the closest one-hot vector representation for the probabilistic prediction of $f_{\mathbf{w}}$, i.e., $R_{\mathcal{S}^{in}}(f_{\mathbf{w}}, \widehat{f}) = \mathbb{E}_{\mathbf{x} \sim \mathcal{S}^{in}} \ell(f_{\mathbf{w}}, \widehat{f})$ and $R_{\mathcal{S}^{c}_{selected}}(f_{\mathbf{w}}, \widehat{f}) = \mathbb{E}_{\mathbf{x} \sim \mathcal{S}^{c}_{selected}} \ell(f_{\mathbf{w}}, \widehat{f})$.*

**Practical implications.** Theorem 1 states that the generalization error of the multi-class classifier is upper bounded. If the sizes of the labeled ID $n$ and the selected covariate OOD data $m_c$ are relatively large, the optimal ID loss $\gamma$ is small, and the optimal risk on covariate OOD $R_{\mathbb{P}^{covariate}_{out}}(f_{\mathbf{w}^{*}})$, then the upper bound will mainly depend on the gradient discrepancy between the ID and covariate OOD data selected by our learning algorithm. Notably, this bound synergistically aligns with our gradient-based score (Equation 5). Empirically, we verify these conditions of Theorem 1 and our assumptions in Appendix M, which can hold in practice.

## 4 Experiments

### 4.1 Settings

**Datasets and benchmarks.** Following the setup in Bai et al. (2023), we employ CIFAR-10 (Krizhevsky et al., 2009) as $\mathbb{P}_{\text{in}}$ and CIFAR-10-C (Hendrycks & Dietterich, 2018) with Gaussian additive noise as the $\mathbb{P}_{\text{out}}^{\text{covariate}}$. For $\mathbb{P}_{\text{out}}^{\text{semantic}}$, we leverage SVHN (Netzer et al., 2011), Textures (Cimpoi et al., 2014), Places365 (Zhou et al., 2017), and LSUN (Yu et al., 2015). We divide CIFAR-10 training set into 50% labeled as ID and 50% unlabeled. And we mix unlabeled CIFAR-10, CIFAR-10-C, and semantic OOD data to generate the wild dataset. To simulate the wild distribution $\mathbb{P}_{\text{wild}}$, we adopt the same mixture ratio as in SCONE (Bai et al., 2023), where $\pi_c = 0.5$ and $\pi_s = 0.1$. Detailed descriptions of the datasets and data mixture can be found in the Appendix B. To demonstrate the adaptability and robustness of our proposed method, we extend the framework to more diverse settings and datasets. Additional results on other types of covariate shifts can be found in Appendix E.

**Experimental details.** To ensure a fair comparison with prior works (Bai et al., 2023; Liu et al., 2020; Katz-Samuels et al., 2022), we adopt Wide ResNet with 40 layers and a widen factor of 2 (Zagoruyko &

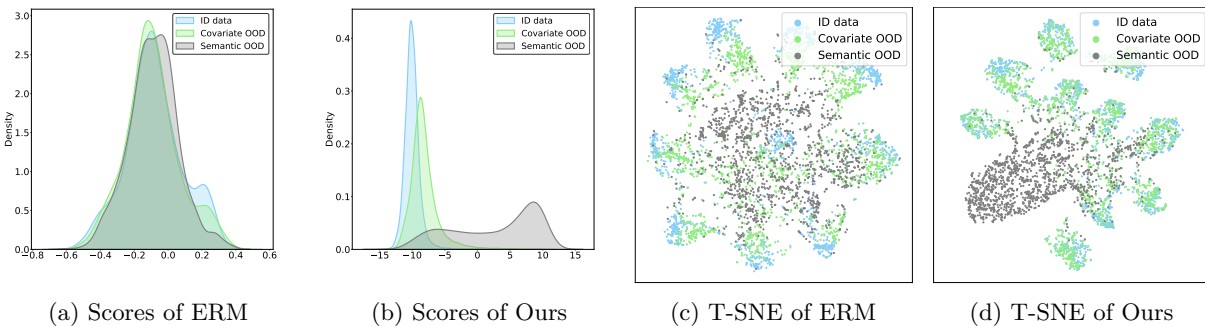

(a) Scores of ERM   (b) Scores of Ours   (c) T-SNE of ERM   (d) T-SNE of Ours

Figure 3: (a)-(b) Score distributions for ERM vs. our method. Different colors represent the different types of test data: CIFAR-10 as $\mathbb{P}_{\text{in}}$ (blue), CIFAR-10-C as $\mathbb{P}_{\text{out}}^{\text{covariate}}$ (green), and Textures as $\mathbb{P}_{\text{out}}^{\text{semantic}}$ (gray). (c)-(d): T-SNE visualization of the image embeddings using ERM vs. our method.

Komodakis, 2016). We use stochastic gradient descent with Nesterov momentum (Duchi et al., 2011), with weight decay 0.0005 and momentum 0.09. The model is initialized with a pre-trained network on CIFAR-10, and then trained for 100 epochs using our objective in Equation 7, with $\alpha = 10$. We use a batch size of 128 and an initial learning rate of 0.1 with cosine learning rate decay. We default $k$ to 1000 and provide analysis of different labeling budgets $k \in \{100, 500, 1000, 2000\}$ in Section 4.3. In our experiment, the output of $g_{\boldsymbol{\theta}}$ is utilized as the score for OOD detection. In practice, we find that using one round of human feedback is sufficient to achieve strong performance. Our implementation is based on PyTorch 1.8.1. All experiments are performed using NVIDIA GeForce RTX 2080 Ti.

**Evaluation metrics.** We report the accuracy of the ID and covariate OOD data, to measure the classification and OOD generalization performance. In addition, we report false positive rate (FPR) and AUROC for the OOD detection performance. The threshold for the OOD detector is selected based on the ID data when 95% of ID test data points are declared as ID.

## 4.2 Main Results

**Competitive performance on both OOD detection and generalization tasks.** As shown in Table 1, our approach achieves strong performance for both OOD generalization and OOD detection tasks jointly. For a comprehensive evaluation, we compare our method with three categories of methods: **(1)** methods developed specifically for OOD detection, **(2)** methods developed specifically for OOD generalization, and **(3)** methods that are trained with wild data like ours. We discuss them below.

First, we observe that our approach achieves superior performance compared to OOD detection baselines, including MSP (Hendrycks & Gimpel, 2016), ODIN (Liang et al., 2018a), Energy (Liu et al., 2020), Mahalanobis (Lee et al., 2018), ViM (Wang et al., 2022b), KNN (Sun et al., 2022), and latest baseline ASH (Djurisic et al., 2023a). Methods tailored for OOD detection tend to capture the domain-variant information and struggle with the covariate distribution shift, resulting in suboptimal OOD accuracy. For example, our method achieves near-perfect FPR95 (0.12%), when evaluating against SVHN as semantic OOD. Secondly, while approaches for OOD generalization, containing IRM (Arjovsky et al., 2019), ERM, Mixup (Zhang et al., 2018), VREx (Krueger et al., 2021), EQRM (Eastwood et al., 2022a), and latest baseline SharpDRO (Huang et al., 2023b), demonstrate improved OOD accuracy, they cannot effectively distinguish between ID data and semantic OOD data, leading to poor OOD detection performance. Lastly, closest to our setting, we compare with strong baselines trained with wild data, namely Outlier Exposure (Hendrycks et al., 2018), Energy-regularized learning (Liu et al., 2020), WOODS (Katz-Samuels et al., 2022), and SCONE (Bai et al., 2023). These methods emerge as robust OOD detectors, yet display a notable decline in OOD generalization (except for SCONE). In contrast, our method demonstrates consistently better results in terms of both OOD generalization and detection performance. Notably, our method even surpasses the current state-of-the-art method SCONE by 32.24% in terms of FPR95 on the Texture OOD dataset, and simultaneously improves the OOD accuracy by 4.75% on CIFAR-10-C.

Table 1: **Main results**: comparison with competitive OOD generalization and detection methods on CIFAR-10. Results for LSUN-R and Texture datasets are in Appendix D. *Since all the OOD detection methods use the same model trained with the CE loss on $\mathbb{P}_{in}$, they display the same ID and OOD accuracy on CIFAR-10-C. We report the average and std of our method based on **3 independent runs**. $\pm x$ denotes the rounded standard error. *Note that OOD Acc. metric refers to the classification accuracy on covariate OOD data while AUROC and FPR are for OOD detection evaluation.*

| Method | SVHN $\mathbb{P}^{semantic}_{out}$, CIFAR-10-C $\mathbb{P}^{covariate}_{out}$ | | | | LSUN-C $\mathbb{P}^{semantic}_{out}$, CIFAR-10-C $\mathbb{P}^{covariate}_{out}$ | | | | Texture $\mathbb{P}^{semantic}_{out}$, CIFAR-10-C $\mathbb{P}^{covariate}_{out}$ | | | |
|---|---|---|---|---|---|---|---|---|---|---|---|---|
| | OOD Acc.↑ | ID Acc.↑ | FPR↓ | AUROC↑ | OOD Acc.↑ | ID Acc.↑ | FPR↓ | AUROC↑ | OOD Acc.↑ | ID Acc.↑ | FPR↓ | AUROC↑ |
| *OOD detection* | | | | | | | | | | | | |
| MSP | 75.05* | 94.84* | 48.49 | 91.89 | 75.05 | 94.84 | 30.80 | 95.65 | 75.05 | 94.84 | 59.28 | 88.50 |
| ODIN | 75.05 | 94.84 | 33.35 | 91.96 | 75.05 | 94.84 | 15.52 | 97.04 | 75.05 | 94.84 | 49.12 | 84.97 |
| Energy | 75.05 | 94.84 | 35.59 | 90.96 | 75.05 | 94.84 | 8.26 | 98.35 | 75.05 | 94.84 | 52.79 | 85.22 |
| Mahalanobis | 75.05 | 94.84 | 12.89 | 97.62 | 75.05 | 94.84 | 39.22 | 94.15 | 75.05 | 94.84 | 15.00 | 97.33 |
| ViM | 75.05 | 94.84 | 21.95 | 95.48 | 75.05 | 94.84 | 5.90 | 98.82 | 75.05 | 94.84 | 29.35 | 93.70 |
| KNN | 75.05 | 94.84 | 28.92 | 95.71 | 75.05 | 94.84 | 28.08 | 95.33 | 75.05 | 94.84 | 39.50 | 92.73 |
| ASH | 75.05 | 94.84 | 40.76 | 90.16 | 75.05 | 94.84 | 2.39 | 99.35 | 75.05 | 94.84 | 53.37 | 85.63 |
| *OOD generalization* | | | | | | | | | | | | |
| ERM | 75.05 | 94.84 | 35.59 | 90.96 | 75.05 | 94.84 | 8.26 | 98.35 | 75.05 | 94.84 | 52.79 | 85.22 |
| Mixup | 79.17 | 93.30 | 91.96 | 18.78 | 79.17 | 93.30 | 52.10 | 76.66 | 79.17 | 93.30 | 58.24 | 75.70 |
| IRM | 77.92 | 90.85 | 63.65 | 90.70 | 77.92 | 90.85 | 36.67 | 94.22 | 77.92 | 90.85 | 59.42 | 87.81 |
| VREx | 76.90 | 91.35 | 55.92 | 91.22 | 76.90 | 91.35 | 51.50 | 91.56 | 76.90 | 91.35 | 65.45 | 85.46 |
| EQRM | 75.71 | 92.93 | 51.86 | 90.92 | 75.71 | 92.93 | 21.53 | 96.49 | 75.71 | 92.93 | 57.18 | 89.11 |
| SharpDRO | 79.03 | **94.91** | 21.24 | 96.14 | 79.03 | 94.91 | 5.67 | 98.71 | 79.03 | **94.91** | 42.94 | 89.99 |
| *Learning w. $\mathbb{P}_{wild}$* | | | | | | | | | | | | |
| OE | 37.61 | 94.68 | 0.84 | 99.80 | 41.37 | 93.99 | 3.07 | 99.26 | 44.71 | 92.84 | 29.36 | 93.93 |
| Energy (w. outlier) | 20.74 | 90.22 | 0.86 | 99.81 | 32.55 | 92.97 | 2.33 | 99.93 | 49.34 | 94.68 | 16.42 | 96.46 |
| Woods | 52.76 | 94.86 | 2.11 | 99.67 | 76.90 | **95.02** | 1.80 | 99.56 | 83.14 | 94.49 | 39.10 | 90.45 |
| Scone | 84.69 | 94.65 | 10.86 | 97.84 | 84.58 | 93.73 | 10.23 | 98.02 | 85.56 | 93.97 | 37.15 | 90.91 |
| **Ours** | **88.26**$_{\pm0.07}$ | 94.68$_{\pm0.07}$ | **0.12**$_{\pm0.00}$ | **99.98**$_{\pm0.00}$ | **90.63**$_{\pm0.02}$ | 94.33$_{\pm0.01}$ | **0.07**$_{\pm0.00}$ | **99.97**$_{\pm0.00}$ | **90.31**$_{\pm0.02}$ | 94.33$_{\pm0.03}$ | **4.91**$_{\pm0.03}$ | **98.28**$_{\pm0.01}$ |

| Algorithm | Art painting | Cartoon | Photo | Sketch | Avg (%) |
|---|---|---|---|---|---|
| **IRM** (Arjovsky et al., 2019) | 84.8 | 76.4 | 96.7 | 76.1 | 83.5 |
| **DANN** (Ganin et al., 2016) | 86.4 | 77.4 | 97.3 | 73.5 | 83.7 |
| **CDANN** (Li et al., 2018c) | 84.6 | 75.5 | 96.8 | 73.5 | 82.6 |
| **GroupDRO** (Sagawa et al., 2020) | 83.5 | 79.1 | 96.7 | 78.3 | 84.4 |
| **MTL** (Blanchard et al., 2021) | 87.5 | 77.1 | 96.4 | 77.3 | 84.6 |
| **I-Mixup** (Wang et al., 2020) | 86.1 | 78.9 | 97.6 | 75.8 | 84.6 |
| **MMD** (Li et al., 2018b) | 86.1 | 79.4 | 96.6 | 76.5 | 84.7 |
| **VREx** (Krueger et al., 2021) | 86.0 | 79.1 | 96.9 | 77.7 | 84.9 |
| **MLDG** (Li et al., 2018a) | 85.5 | 80.1 | 97.4 | 76.6 | 84.9 |
| **ARM** (Zhang et al., 2021b) | 86.8 | 76.8 | 97.4 | 79.3 | 85.1 |
| **RSC** (Huang et al., 2020) | 85.4 | 79.7 | 97.6 | 78.2 | 85.2 |
| **Mixstyle** (Zhou et al., 2021) | 86.8 | 79.0 | 96.6 | 78.5 | 85.2 |
| **ERM** (Vapnik, 1999) | 84.7 | 80.8 | 97.2 | 79.3 | 85.5 |
| **CORAL** (Sun & Saenko, 2016) | 88.3 | 80.0 | 97.5 | 78.8 | 86.2 |
| **SagNet** (Nam et al., 2021) | 87.4 | 80.7 | 97.1 | 80.0 | 86.3 |
| **SelfReg** (Kim et al., 2021) | 87.9 | 79.4 | 96.8 | 78.3 | 85.6 |
| **GVRT** Min et al. (2022) | 87.9 | 78.4 | 98.2 | 75.7 | 85.1 |
| **VNE** (Kim et al., 2023) | **88.6** | 79.9 | 96.7 | 82.3 | 86.9 |
| **Ours** | 88.1 | **87.4** | **98.5** | **91.3** | **91.3** |

Table 2: Comparison with domain generalization methods on the PACS benchmark. All methods are trained on ResNet-50. The model selection is based on a training domain validation set.

**Additional results on PACS benchmark.** In Table 2, we report results on the PACS dataset (Li et al., 2017) from DomainBed. We compare our approach with various common OOD generalization baselines, including IRM (Arjovsky et al., 2019), DANN (Ganin et al., 2016), CDANN (Li et al., 2018c), Group-DRO (Sagawa et al., 2020), MTL (Blanchard et al., 2021), I-Mixup (Wang et al., 2020), MMD (Li et al., 2018b), VREx (Krueger et al., 2021), MLDG (Li et al., 2018a), ARM (Zhang et al., 2021b), RSC (Huang et al., 2020), Mixstyle (Zhou et al., 2021), ERM (Vapnik, 1999), CORAL (Sun & Saenko, 2016), SagNet (Nam et al., 2021), SelfReg (Kim et al., 2021), GVRT (Min et al., 2022), and the latest baseline VNE (Kim et al., 2023). Our method achieves an average accuracy of 91.3%, which outperforms these OOD generalization baselines.

**Visualization of OOD score distributions.** Figure 3 (a) and (b) visualize the score distributions for ERM (without human feedback) vs. our method. The OOD score distributions between $\mathbb{P}_{in}$ and $\mathbb{P}^{semantic}_{out}$ are more clearly differentiated using our method. This separation leads to an improvement in OOD detection

performance. The enhanced separation can be attributed to the effectiveness of the OOD learning with human feedback framework in recognizing semantic-shifted OOD data.

**Visualization of feature embeddings.** Figure 3 (c) and (d) present t-SNE visualizations (Van der Maaten & Hinton, 2008) of feature embeddings on the test data. The blue points represent the test ID data (CIFAR-10), green points denote test samples from CIFAR-10-C, and gray points are from the Texture dataset. This visualization indicates that embeddings of $\mathbb{P}_{\text{in}}$ (CIFAR) and $\mathbb{P}_{\text{out}}^{\text{covariate}}$ (CIFAR-C) are more closely aligned using our method, which contributes to enhanced OOD generalization performance.

### 4.3 In-Depth Analysis

**Impact of labeling budget $k$.** The budget $k$ is central to our OOD learning with human feedback framework. In table 3, we conduct an ablation by varying $k \in \{100, 500, 1000, 2000\}$. We observe that the performance of OOD generalization and OOD detection both increase with a larger number of annotation budgets. For example, The OOD accuracy improves from 86.62% to 91.09% when the budget changes from $k = 100$ to $k = 2000$. At the same

Table 3: Ablation on labeling budget $k$. We train on CIFAR-10 as ID, using wild data with $\pi_c = 0.5$ (CIFAR-10-C) and $\pi_s = 0.1$ (Texture).

| Budget $k$ | OOD Acc.↑ | ID Acc.↑ | FPR↓ | AUROC↑ |
|---|---|---|---|---|
| 100 | 86.62 | 95.03 | 22.16 | 91.34 |
| 500 | 89.96 | 94.70 | 7.50 | 97.45 |
| 1000 | 90.31 | 94.33 | 4.91 | 98.28 |
| 2000 | 91.09 | 94.42 | 3.60 | 98.90 |

time, the FPR95 reduces from 22.16% to 3.60%. Interestingly, we do notice a marginal difference between $k = 1000$ and $k = 2000$, which suggests that our method suffices to achieve strong performance without excessive labeling budget.

**Impact of different sampling scores.** Our main results are based on the gradient-based sampling score (*c.f.* Section 3.1). Here we provide additional comparison using different sampling scores, including least-confidence sampling (Wang & Shang, 2014), entropy-based sampling (Wang & Shang, 2014), margin-based sampling (Roth & Small, 2006), energy score (Liu et al., 2020), BADGE (Ash et al., 2019), and random sampling. Detailed description for each method is provided in Appendix C. We observe that the gradient-based score demonstrates overall strong performance in terms of OOD generalization and detection.

Table 4: Impact of sampling scores. We use budget $k = 1000$ for all methods. We train on CIFAR-10 as ID, using wild data with $\pi_c = 0.5$ (CIFAR-10-C) and $\pi_s = 0.1$ (Texture).

| Sampling score | OOD Acc.↑ | ID Acc.↑ | FPR↓ | AUROC↑ |
|---|---|---|---|---|
| Least confidence | 90.22 | 94.73 | 8.44 | 96.45 |
| Entropy | 90.02 | 94.80 | 8.19 | 96.66 |
| Margin | 90.31 | 94.56 | 6.37 | 97.68 |
| BADGE | 88.68 | 94.56 | 8.77 | 96.39 |
| Random | 89.22 | **94.84** | 9.45 | 95.41 |
| Energy score | 88.75 | 94.78 | 10.89 | 95.19 |
| Gradient-based | **90.31** | 94.33 | **4.91** | **98.28** |

**Impact of different sampling strategies.** In Table 5, we compare the performance of using three different sampling strategies (1) top-$k$ sampling, (2) near-boundary sampling, and (3) mixed sampling. For all three strategies, we employ the same labeling budget $k = 100$ and the same gradient-based scoring function (*c.f.* Section 3.1). We observe that the top-$k$ sampling achieves the best OOD generalization performance. This is because the selected samples are furthest away from the ID data, presenting challenging cases of covariate-shifted OOD data. By labeling and learning from these hard cases, the multi-class classifier acquires a stronger generalization to OOD data. Moreover, near-boundary sampling displays the lowest performance in both OOD generalization and OOD detection. To reason for this, we provide the number of samples (among 100) belonging to ID, covariate OOD, and semantic OOD respectively. As seen in Table 5, the majority of the samples appear to be either ID or covariate OOD (easy cases), whereas only 6 out of 100 samples are semantic OOD. As a result, this sampling strategy does not provide sufficient informative samples needed for the OOD detector. Lastly, the mixed sampling strategy achieves performance somewhere in between top-$k$ sampling and near-boundary sampling, which aligns with our expectations.

Table 5: Impact of sampling strategy ($k = 100$). We train on CIFAR-10 as ID, using wild data with $\pi_c = 0.5$ (CIFAR-10-C) and $\pi_s = 0.1$ (Texture).

| Sampling Strategies | OOD Acc.↑ | ID Acc.↑ | FPR↓ | AUROC↑ | #ID | #Covariate OOD | #Semantic OOD |
|---|---|---|---|---|---|---|---|
| Top-$k$ sampling | 86.62 | 95.03 | 22.16 | 91.34 | 0 | 57 | 43 |
| Near-boundary sampling | 85.12 | 95.18 | 41.72 | 76.56 | 44 | 50 | 6 |
| Mixed sampling | 85.36 | 95.10 | 36.24 | 85.37 | 25 | 51 | 24 |

## 5 Related Works

**Out-of-distribution generalization** is a crucial problem in machine learning when training and test data are sampled from different distributions. Compared to domain adaptation task (You et al., 2019; Kumar et al., 2020; Wang et al., 2022c; Prabhu et al., 2021; Su et al., 2020; Kothandaraman et al., 2023), OOD generalization is more challenging, as it focuses on adapting to *unseen* covariate-shifted data without access to any sample from the target domain (Gulrajani & Lopez-Paz, 2020; Bai et al., 2021; Koh et al., 2021; Ye et al., 2022; Cho et al., 2023; Bai et al., 2024a;b). Existing theoretical work on domain adaptation (Mansour et al., 2009; Blitzer et al., 2007; Ben-David et al., 2006; 2010; Gui et al., 2024) discusses the generalization error bound with access to target domain data. Our analysis presents several key distinctions: (1) Our focus is on the wild setting, necessitating an additional step of selection and human annotation to acquire selected covariate OOD data for retraining; (2) Our OOD generalization error bound differs in that it is based on a gradient-based discrepancy between ID and OOD data. This diverges from classical domain adaptation literature and synergistically aligns with our gradient-based sampling score.

A prevalent approach in the OOD generalization area is to learn a domain-invariant data representation across training domains. This involves various strategies like invariant risk minimization (Arjovsky et al., 2019; Ahuja et al., 2020; Krueger et al., 2021; Eastwood et al., 2022b), robust optimization (Sagawa et al., 2020; Dai et al., 2023; Huang et al., 2023a), domain adversarial learning (Li et al., 2018b; Wang et al., 2022d; Dayal et al., 2023), meta-learning (Li et al., 2018a; Zhang et al., 2021a), and gradient alignment (Shi et al., 2021; Rame et al., 2022; Guo et al., 2023). Some OOD algorithms do not require multiple training domains (Tong et al., 2023). Other approaches include model ensembles (Chen et al., 2023c; Ramé et al., 2023), graph learning (Gui et al., 2023; Yuan et al., 2023), and test-time adaptation (Chen et al., 2023a; Park et al., 2023; Samadh et al., 2023; Chen et al., 2023b). SCONE (Bai et al., 2023) aims to enhance OOD robustness and detection by utilizing wild data from the open world. Building on SCONE's problem setting, we leverage human feedback to train a robust classifier and OOD detector, supported by theoretical analysis.

**Out-of-distribution detection** has garnered increasing interest in recent years (Yang et al., 2021; 2022; Zhang et al., 2023b; Du et al., 2024b). Recent methods can be broadly classified into post hoc and regularized-based algorithms. Post hoc methods, which include confidence-based methods (Hendrycks & Gimpel, 2016; Liang et al., 2018b), energy-based scores (Liu et al., 2020; Wang et al., 2021; Sun et al., 2021; Djurisic et al., 2023b; Zhang et al., 2023c; Lafon et al., 2023), gradient-based score (Huang et al., 2021; Behpour et al., 2023), Bayesian approaches (Gal & Ghahramani, 2016; Maddox et al., 2019; Kristiadi et al., 2020), and distance-based methods (Lee et al., 2018; Sun et al., 2022; Du et al., 2021; 2022a; Ming et al., 2022b), perform OOD detection by devising OOD scores at test time. On the other hand, another line of work addresses OOD detection through training-time regularization (Hendrycks et al., 2018; Hein et al., 2019; Du et al., 2022c;b; Ming et al., 2022a; Wang et al., 2023; Du et al., 2023; Tao et al., 2023), which typically relies on a clean set of auxiliary semantic OOD data. WOODS (Katz-Samuels et al., 2022) and SAL (Du et al., 2024a) address this issue by utilizing wild mixture data, comprising both unlabeled ID and semantic OOD data. While SAL also employs a gradient-based score, it does not consider OOD generalization or leveraging human feedback, which is our main focus. Our work builds upon the setting in SCONE (Bai et al., 2023) and introduces a novel OOD learning with human feedback framework aimed at enhancing both OOD generalization and detection jointly.

**Active learning** emphasizes the selection of the most informative data points for labeling (Cohn et al., 1994; Balcan et al., 2006; Settles, 2009; Wang & Shang, 2014; Ren et al., 2021; Karzand & Nowak, 2020; Xie et al., 2024). Well-known sampling strategies include disagreement-based sampling, diversity sampling, and uncertainty sampling. Disagreement-based sampling (Seung et al., 1992; Hanneke et al., 2014; Zhu & Nowak, 2022) focuses on selecting data points that elicit disagreement among multiple models. Diversity

sampling, as explored in (Du et al., 2015; Zhdanov, 2019; Citovsky et al., 2021), aims to select data points that are both diverse and representative of the data's overall distribution. Uncertainty sampling (Lewis, 1995; Scheffer et al., 2001; Shannon, 2001; Lu et al., 2016; Wang & Shang, 2014; Ducoffe & Precioso, 2018; Beluch et al., 2018) seeks to identify data points where model confidence is lowest, thus reducing uncertainty. More advanced methods (Ash et al., 2019; Wang et al., 2022a; Elenter et al., 2022; Mohamadi et al., 2022) incorporate a mix of uncertainty and diversity sampling techniques. Another research direction involves deep active learning under data imbalance (Kothawade et al., 2021; Emam et al., 2021; Zhang et al., 2022; Coleman et al., 2022; Zhang et al., 2023a; Kothawade et al., 2022; Aggarwal et al., 2020). The work of (Das et al., 2023; Deng et al., 2023; Zhan et al., 2023; Shayovitz et al., 2024; Benkert et al., 2022) considers distribution shifts in the context of active learning. However, previous approaches usually focus on selecting informative samples for improving ID classification, which do not consider OOD robustness and the challenges posed by realistic scenarios involving wild data. In our work, we introduce a novel framework tailored for both OOD generalization and detection challenges. This framework is further supported by a theoretical justification of our learning approach.

## 6    Conclusion

This paper introduces a new framework leveraging human feedback for both OOD generalization and OOD detection. Our framework tackles the fundamental challenge of leveraging the wild data—the lack of supervision for samples from the wild distribution. Specifically, we employ a gradient-based sampling score to selectively label informative OOD samples from the wild data distribution and then train a robust multi-class classifier and an OOD detector. Importantly, our algorithm only requires a small annotation budget and performs competitively compared to various baselines, which offers practical advantages. We further provide theoretical analysis of the learnability of the classifier. We hope our work will inspire future research on both empirical and theoretical understanding of OOD generalization and detection in a synergistic way.

## 7    Acknowledgement

This work is supported by the Office of Naval Research under grant number N00014-23-1-2643, AFOSR Young Investigator Program under award number FA9550-23-1-0184, National Science Foundation (NSF) Award No. IIS-2237037 and IIS-2331669. The authors would also like to thank TMLR reviewers for the helpful suggestions and feedback.

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

# Out-of-Distribution Learning with Human Feedback (Appendix)

## A   Algorithm

---

**Algorithm 1** Out-of-Distribution Learning with Human Feedback

---

**Input:** In-distribution labeled data $\mathcal{S}^{\text{in}} = \{(\mathbf{x}_i, y_i)\}_{i=1}^n$. Unlabeled wild data $\mathcal{S}_{\text{wild}} = \{\tilde{\mathbf{x}}_i\}_{i=1}^m$.
**Output:** Learned classifier $f_{\widehat{\mathbf{w}}}$ and OOD detector $g_{\widehat{\boldsymbol{\theta}}}$.
**Sample section**
1. Perform ERM on labeled ID data $\mathcal{S}^{\text{in}}$ and obtain learned weight parameter $\mathbf{w}_{\mathcal{S}^{\text{in}}}$ according to Eq. equation 2.
2. Calculate the reference gradient $\bar{\nabla}$ on $\mathcal{S}^{\text{in}}$ according to Eq. equation 3.
3. Generate predicted labels $\widehat{y}_{\tilde{\mathbf{x}}_i}$ for $\tilde{\mathbf{x}}_i \in \mathcal{S}_{\text{wild}}$.
4. Calculate gradient $\nabla \ell(h_{\mathbf{w}_{\mathcal{S}^{\text{in}}}}(\tilde{\mathbf{x}}_i), \widehat{y}_{\tilde{\mathbf{x}}_i})$.
5.  Calculate the gradient matrix $\boldsymbol{G}$ by Eq. equation 4 and Compute the gradient-based score $\tau_i$ by Eq. equation 5.
6. Given the gradient-based scores $\tau_i$ for each sample $\tilde{\mathbf{x}}_i$, select a subset $k$ according to the three sampling strategies described in Sec. 3.1.
7. Annotate the selected $k$ samples with labels from: $\mathcal{Y} \cup \{\bot\}$ to obtain $\mathcal{S}^{\text{c}}_{\text{selected}}$ and $\mathcal{S}^{\text{s}}_{\text{selected}}$, where $\{\bot\}$ indicates the semantic OOD.
**OOD learning with annotated samples**
8. Train the robust classifier using samples from $\mathcal{S}^{\text{in}}$ and covariate OOD $\mathcal{S}^{\text{c}}_{\text{selected}}$. Concurrently, train a binary OOD detector using semantic OOD $\mathcal{S}^{\text{s}}_{\text{selected}}$ and $\mathcal{S}^{\text{in}}$ by Eq. equation 7.

---

## B   Detailed Description of Datasets

**CIFAR-10** (Krizhevsky et al., 2009) contains $60,000$ color images with 10 classes. The training set has $50,000$ images and the test set has $10,000$ images.

**CIFAR-10-C** is algorithmically generated, following the previous leterature (Hendrycks & Dietterich, 2018). The corruptions include Gaussian noise, defocus blur, glass blur, impulse noise, shot noise, snow, and zoom blur.

**SVHN** (Netzer et al., 2011) is a real-world image dataset obtained from house numbers in Google Street View images, with 10 classes. This dataset contains $73,257$ samples for training and $26,032$ samples for testing.

**Places365** (Zhou et al., 2017) contains scene photographs and diverse types of environments encountered in the world. The scene semantic categories consist of three macro-classes: Indoor, Nature, and Urban.

**LSUN-C** (Yu et al., 2015) and **LSUN-R** (Yu et al., 2015) are large-scale image datasets that are annotated using deep learning with humans in the loop. LSUN-C is a cropped version of LSUN and LSUN-R is a resized version of the LSUN dataset.

**Textures** (Cimpoi et al., 2014) refers to the Describable Textures Dataset, which contains images of patterns and textures. The subset we used has no overlap categories with the CIFAR dataset (Krizhevsky et al., 2009).

**PACS** (Li et al., 2017) is commonly used in evaluating OOD generalization approaches. This dataset consists of $9,991$ examples of resolution $224 \times 224$ and four domains with different image styles including photo, art painting, cartoon, and sketch with seven categories.

**Details of data split for OOD datasets.** For datasets with standard train-test split (e.g., SVHN), we use the original test split for evaluation. For other OOD datasets (e.g., LSUN-C), we use 70% of the data for creating the wild mixture training data as well as the mixture validation dataset. We use the

remaining examples for test-time evaluation. For splitting training/validation, we use 30% for validation and the remaining for training.

## C Description of Sampling Methods

**Least confidence** (Wang & Shang, 2014) is an uncertainty-based algorithm that selects data for which the most probable label possesses the lowest posterior probability. This method focuses on instances where the model's predictions are least certain.

**Margin sampling** (Roth & Small, 2006) elects data points based on the multiclass margin, specifically targeting examples where the posterior probabilities of the two most likely labels are closely matched, indicating the minimal difference between them.

**Entropy** (Wang & Shang, 2014) is an uncertainty-based algorithm that chooses data points by evaluating the entropy within the predictive class probability distribution of each example, aiming to maximize the overall predictive entropy.

**Energy sampling** (Liu et al., 2020) identifies data points using an energy score, a measure theoretically aligned with the probability density of the inputs.

**BADGE** (Ash et al., 2019) samples groups of points that are both diverse and exhibit high magnitude in a hallucinated gradient space. This technique combines predictive uncertainty and sample diversity, enhancing the effectiveness of data selection in active learning.

**Random sampling** serves as a straightforward baseline method, involving the random selection of $k$ examples to query.

## D Results on Additional OOD Datasets

In this section, we provide the main results on more OOD datasets including Places365 (Zhou et al., 2017) and LSUN-R (Yu et al., 2015). As shown in Table 6. our proposed approach achieves overall strong performance in OOD generalization and OOD detection on these additional OOD datasets. Firstly, we compare our method with post hoc OOD detection methods such as MSP (Hendrycks & Gimpel, 2016), ODIN (Liang et al., 2018a), Energy (Liu et al., 2020), Mahalanobis (Lee et al., 2018), ViM (Wang et al., 2022b), KNN (Sun et al., 2022), and latest baseline ASH (Djurisic et al., 2023a). These methods are all based on a model trained with cross-entropy loss, which suffers from limiting OOD generalization performance. However, our method achieves an improved OOD generalization performance (e.g., 91.08% when the wild data is a mixture of CIFAR-10, CIFAR-10-C, and LSUN-R).

Secondly, we also compare our method with common OOD generalization baseline methods including IRM (Arjovsky et al., 2019), ERM, Mixup (Zhang et al., 2018), VREx (Krueger et al., 2021), EQRM (Eastwood et al., 2022a), and latest baseline SharpDRO (Huang et al., 2023b). Our approach consistently achieves better results compared to these OOD generalization baselines. Lastly, we compare our method with strong OOD baselines using $\mathbb{P}_{\text{wild}}$ such as Outlier Exposure (Hendrycks et al., 2018), Energy-regularized learning (Liu et al., 2020), WOODS (Katz-Samuels et al., 2022), and SCONE (Bai et al., 2023). Contrastly, our approach demonstrates strong performance on both OOD generalization and detection accuracy, which shows the effectiveness of our method for making use of the wild data.

## E Results on Different Corruption Types

In this section, we provide additional ablation studies of the different covariate shifts. In Table 7, we evaluate our method under different common corruptions including Gaussian noise, shot noise, glass blur, and etc. To generate images with corruption, we follow the default setting and hyperparameters as in Hendrycks & Dietterich (2018). Our approach is robust under different covariate shifts and achieves strong OOD detection performance.

Table 6: Additional results. Comparison with competitive OOD detection and OOD generalization methods on CIFAR-10. For experiments using $\mathbb{P}_{\text{wild}}$, we use $\pi_s = 0.5$, $\pi_c = 0.1$. For each semantic OOD dataset, we create corresponding wild mixture distribution $\mathbb{P}_{\text{wild}} := (1 - \pi_s - \pi_c)\mathbb{P}_{\text{in}} + \pi_s \mathbb{P}_{\text{out}}^{\text{semantic}} + \pi_c \mathbb{P}_{\text{out}}^{\text{covariate}}$ for training. We report the average and std of our method based on **3 independent runs**. $\pm x$ denotes the rounded standard error.

| Model | Places365 $\mathbb{P}_{\text{out}}^{\text{semantic}}$, CIFAR-10-C $\mathbb{P}_{\text{out}}^{\text{covariate}}$ | | | | LSUN-R $\mathbb{P}_{\text{out}}^{\text{semantic}}$, CIFAR-10-C $\mathbb{P}_{\text{out}}^{\text{covariate}}$ | | | |
| | OOD Acc.↑ | ID Acc.↑ | FPR↓ | AUROC↑ | OOD Acc.↑ | ID Acc.↑ | FPR↓ | AUROC↑ |
|---|---|---|---|---|---|---|---|---|
| *OOD detection* | | | | | | | | |
| **MSP** | 75.05 | 94.84 | 57.40 | 84.49 | 75.05 | 94.84 | 52.15 | 91.37 |
| **ODIN** | 75.05 | 94.84 | 57.40 | 84.49 | 75.05 | 94.84 | 26.62 | 94.57 |
| **Energy** | 75.05 | 94.84 | 40.14 | 89.89 | 75.05 | 94.84 | 27.58 | 94.24 |
| **Mahalanobis** | 75.05 | 94.84 | 68.57 | 84.61 | 75.05 | 94.84 | 42.62 | 93.23 |
| **ViM** | 75.05 | 94.84 | 21.95 | **95.48** | 75.05 | 94.84 | 36.80 | 93.37 |
| **KNN** | 75.05 | 94.84 | 42.67 | 91.07 | 75.05 | 94.84 | 29.75 | 94.60 |
| **ASH** | 75.05 | 94.84 | 44.07 | 88.84 | 75.05 | 94.84 | 22.07 | 95.61 |
| *OOD generalization* | | | | | | | | |
| **ERM** | 75.05 | 94.84 | 40.14 | 89.89 | 75.05 | 94.84 | 27.58 | 94.24 |
| **Mixup** | 79.17 | 93.30 | 58.24 | 75.70 | 79.17 | 93.30 | 32.73 | 88.86 |
| **IRM** | 77.92 | 90.85 | 53.79 | 88.15 | 77.92 | 90.85 | 34.50 | 94.54 |
| **VREx** | 76.90 | 91.35 | 56.13 | 87.45 | 76.90 | 91.35 | 44.20 | 92.55 |
| **EQRM** | 75.71 | 92.93 | 51.00 | 88.61 | 75.71 | 92.93 | 31.23 | 94.94 |
| **SharpDRO** | 79.03 | **94.91** | 34.64 | 91.96 | 79.03 | 94.91 | 13.27 | 97.44 |
| *Learning w. $\mathbb{P}_{wild}$* | | | | | | | | |
| **OE** | 35.98 | 94.75 | 27.02 | 94.57 | 46.89 | 94.07 | 0.7 | 99.78 |
| **Energy (w/ outlier)** | 19.86 | 90.55 | 23.89 | 93.60 | 32.91 | 93.01 | 0.27 | 99.94 |
| **Woods** | 54.58 | 94.88 | 30.48 | 93.28 | 78.75 | **95.01** | 0.60 | 99.87 |
| **Scone** | 85.21 | 94.59 | 37.56 | 90.90 | 80.31 | 94.97 | 0.87 | 99.79 |
| **Ours** | **89.16**$_{\pm0.01}$ | 94.51$_{\pm0.11}$ | **15.70**$_{\pm0.02}$ | 94.68$_{\pm0.03}$ | **91.08**$_{\pm0.01}$ | 94.41$_{\pm0.00}$ | **0.07**$_{\pm0.00}$ | **99.98**$_{\pm0.00}$ |

Table 7: Ablation study on different corruption types for covariate OOD data. The budget is 500 for active learning. We train on CIFAR-10 as ID, using wild data with $\pi_c = 0.5$ (CIFAR-10-C) and $\pi_s = 0.1$ (Texture).

| Corruption types for covariate OOD data | OOD Acc.↑ | ID Acc.↑ | FPR↓ | AUROC↑ |
|---|---|---|---|---|
| Gaussian noise | 90.31 | 94.33 | 4.91 | 98.28 |
| Defocus blur | 94.54 | 94.68 | 1.38 | 99.56 |
| Frosted glass blur | 82.22 | 94.45 | 8.35 | 96.87 |
| Impulse noise | 91.82 | 94.38 | 3.61 | 98.88 |
| Shot noise | 91.98 | 94.62 | 3.79 | 98.82 |
| Snow | 92.51 | 94.45 | 2.89 | 99.04 |
| Zoom blur | 92.31 | 94.65 | 3.31 | 98.71 |
| Brightness | 94.59 | 94.53 | 1.56 | 99.48 |
| Elastic transform | 91.12 | 94.35 | 2.70 | 99.09 |
| Contrast | 94.15 | 94.64 | 1.56 | 99.50 |
| Fog | 94.34 | 94.67 | 1.26 | 99.53 |
| Forst | 92.42 | 94.49 | 3.19 | 98.79 |
| Gaussian blur | 94.40 | 94.69 | 1.14 | 99.58 |
| Jpeg compression | 89.75 | 94.52 | 3.37 | 98.95 |
| Motion blur | 92.44 | 94.38 | 2.89 | 99.14 |
| Pixelate | 93.08 | 94.42 | 2.28 | 99.24 |
| Saturate | 93.14 | 94.43 | 2.10 | 99.34 |
| Spatter | 93.66 | 94.60 | 1.98 | 99.35 |
| Speckle noise | 92.19 | 94.37 | 3.79 | 98.55 |

# F  Ablations on Mixing Rate $\lambda$

In the mixed sampling strategy, samples are selected using a combination of the top-$k$ and near-boundary sampling methods. We introduce a mixing rate $\lambda$ to determine the composition of the selected samples, where $0 \leq \lambda \leq 1$. Specifically, $k$ samples are chosen from $\mathcal{S}_{\text{wild}}$. This set consists of $k_1$ samples from the top-$k$ method and $k_2$ samples from the near-boundary sampling, where $k = k_1 + k_2$ and $\lambda = \frac{k_1}{k_1 + k_2}$. In Table 8, we perform ablation on how $\lambda$ affects the performance. We observe that a higher $\lambda$ generally results

Table 8: Ablation results on different mixing rate $\lambda$. The total labeling budget is $k = 500$. We train on CIFAR-10 as ID, using wild data with $\pi_c = 0.5$ (CIFAR-10-C) and $\pi_s = 0.1$ (Texture).

| Mixing rate | OOD Acc.↑ | ID Acc.↑ | FPR↓ | AUROC↑ |
|---|---|---|---|---|
| $\lambda$=0.1 | 87.93 | 94.90 | 16.08 | 92.17 |
| $\lambda$=0.3 | 88.76 | 94.86 | 13.05 | 94.35 |
| $\lambda$=0.5 | 88.08 | 95.02 | 12.98 | 94.41 |
| $\lambda$=0.7 | 89.73 | 94.76 | 9.36 | 96.68 |
| $\lambda$=0.9 | 89.80 | 94.70 | 8.05 | 97.91 |

in stronger performance in both OOD generalization and OOD detection. The observation aligns with our expectations and is supported by the detailed quantitative analysis presented in Section 4.3.

## G Hyperparameter Analysis

In Table 9, we present the performance of OOD generalization and detection by varying hyperparameter $\alpha$, which balances the weight between two loss terms. We observe that the generalization performance remains competitive and insensitive across a wide range of $\alpha$ values. Additionally, our method demonstrates enhanced OOD detection performance when a relatively larger value of $\alpha$ is employed in this scenario.

Table 9: Ablation study on the effect of loss weight $\alpha$. The sampling strategy is top-$k$ sampling, with a budget of 1000. We train on CIFAR-10 as ID, using wild data with $\pi_c = 0.5$ (CIFAR-10-C) and $\pi_s = 0.1$ (Texture).

| Balancing weights | OOD Acc.↑ | ID Acc.↑ | FPR↓ | AUROC↑ |
|---|---|---|---|---|
| $\alpha$=1.0 | 90.76 | 94.49 | 5.29 | 98.33 |
| $\alpha$=3.0 | 90.61 | 94.43 | 5.35 | 98.19 |
| $\alpha$=5.0 | 90.52 | 94.41 | 5.29 | 98.16 |
| $\alpha$=7.0 | 90.51 | 94.35 | 5.41 | 98.15 |
| $\alpha$=9.0 | 90.41 | 94.33 | 5.11 | 98.24 |
| $\alpha$=10.0 | 90.31 | 94.33 | 4.91 | 98.28 |

## H Application to Foundation Models

OOD learning, especially OOD generalization remains a significant challenge even for powerful foundation models like CLIP (Radford et al., 2021), despite their large-scale pretraining. While these models demonstrate impressive zero-shot recognition, they are not inherently robust to all types of OOD shifts. Recent studies (Zhou et al., 2022c;b) have shown that CLIP can still struggle with domain shifts (e.g., changes in lighting, texture, or style) and semantic shifts (e.g., entirely novel categories outside its training distribution). Moreover, spurious correlations learned during large-scale training can degrade performance when encountering unseen real-world variations.

Unlike task-specific models like Wide ResNet trained on CIFAR-10, foundation models are deployed in highly open-ended environments, where the distribution of encountered data is constantly evolving. This makes selective human feedback and adaptive learning strategies, like those proposed in our framework, crucial for improving reliability. Our method can be integrated with CLIP or other foundation models by identifying and adapting to OOD instances dynamically, ensuring that even large-scale models remain robust and trustworthy in real-world deployments, e.g., biomedical assisting systems for question answering and image analysis. Thus, OOD generalization is still a critical research problem, even for state-of-the-art vision foundation models.

# I Notations, Definitions, and Assumptions

Here we summarize important notations in Table 10, restate necessary definitions and assumptions in Sections I.2 and I.3.

## I.1 Notations

Please see Table 10 for detailed notations.

Table 10: Main notations and their descriptions.

| Notation | Description |
|---|---|
| *Spaces* | |
| $\mathcal{X}, \mathcal{Y}$ | the input space and the label space. |
| $\mathcal{W}, \Theta$ | the hypothesis spaces |
| *Distributions* | |
| $\mathbb{P}_{\text{wild}}, \mathbb{P}_{\text{in}}$ | data distribution for wild data, labeled ID data. |
| $\mathbb{P}_{\text{out}}^{\text{covariate}}$ | data distribution for covariate-shifted OOD data. |
| $\mathbb{P}_{\text{out}}^{\text{semantic}}$ | data distribution for semantic-shifted OOD data. |
| $\mathbb{P}_{\mathcal{X}\mathcal{Y}}$ | the joint data distribution for ID data. |
| *Data and Models* | |
| $\mathbf{w}, \mathbf{x}, \mathbf{v}$ | weight/input/the top-1 right singular vector of $\mathbf{G}$ |
| $\widehat{\nabla}, \tau$ | the average gradients on labeled ID data, uncertainty score |
| $\mathcal{S}^{\text{in}}, \mathcal{S}_{\text{wild}}$ | labeled ID data and unlabeled wild data |
| $\mathcal{S}_{\text{selected}}$ | selected data |
| $\mathcal{S}_{\text{selected}}^{\text{s}}, \mathcal{S}_{\text{selected}}^{\text{c}}$ | semantic and covariate OOD in the selected data $\mathcal{S}_{\text{selected}}$ |
| $f_{\mathbf{w}}$ and $g_{\boldsymbol{\theta}}$ | predictor on labeled in-distribution and binary predictor for OOD detection |
| $y$ | label for ID classification |
| $\widehat{y}_{\mathbf{x}}$ | Predicted one-hot label for input $\mathbf{x}$ |
| $n, m, k$ | size of $\mathcal{S}^{\text{in}}$, size of $\mathcal{S}_{\text{wild}}$, labeling budget |
| *Distances* | |
| $d_{\mathcal{W} \triangle \mathcal{W}}(\cdot, \cdot)$ | $\mathcal{W} \triangle \mathcal{W}$ distance. |
| $r_1$ and $r_2$ | the radius of the hypothesis spaces $\mathcal{W}$ and $\Theta$, respectively |
| $\|\cdot\|_2$ | $\ell_2$ norm |
| *Loss, Risk and Predictor* | |
| $\ell(\cdot, \cdot)$ | ID loss function |
| $R_{\mathcal{S}^{\text{in}}, \mathcal{S}_{\text{selected}}^{\text{s}}}(g_{\boldsymbol{\theta}})$ | the overall empirical risk that classifies ID and detects semantic OOD |
| $R_{\mathcal{S}^{\text{in}}, \mathcal{S}_{\text{selected}}^{\text{c}}}(f_{\mathbf{w}})$ | the overall empirical risk that classifies covariate OOD and ID |
| $R_{\mathcal{S}^{\text{in}}}(f_{\mathbf{w}})$ | the empirical risk w.r.t. predictor $f_{\mathbf{w}}$ over data $\mathcal{S}^{\text{in}}$ |
| $R_{\mathcal{S}_{\text{selected}}^{\text{c}}}(f_{\mathbf{w}})$ | the empirical risk w.r.t. predictor $f_{\mathbf{w}}$ over covariate OOD $\mathcal{S}_{\text{selected}}^{\text{c}}$ |
| ID-Acc | in-distribution accuracy. |
| OOD-Acc | out-of-distribution accuracy. |
| FPR | OOD detection performance. |
| *Additional Notations in Theory* | |
| $\omega_{\text{in}}, \omega_{\text{c}}$ | the weight coefficients for ID empirical risk, and covariate OOD empirical risk. |
| $M = \beta_1 r_1^2 + b_1 r_1 + B_1$ | the upper bound of loss $\ell(\mathbf{h}_{\mathbf{w}}(\mathbf{x}), y)$, see Proposition 3 |
| $d$ | VC dimension of the hypothesis space $\mathcal{W}$ |

## I.2 Definitions

**Definition 2** ($\beta$-smooth). *We say a loss function $\ell(f_{\mathbf{w}}(\mathbf{x}), y)$ (defined over $\mathcal{X} \times \mathcal{Y}$) is $\beta$-smooth, if for any $\mathbf{x} \in \mathcal{X}$ and $y \in \mathcal{Y}$,*

$$\left\| \nabla \ell(f_{\mathbf{w}}(\mathbf{x}), y) - \nabla \ell(f_{\mathbf{w}'}(\mathbf{x}), y) \right\|_2 \leq \beta \|\mathbf{w} - \mathbf{w}'\|_2$$

**Definition 3** ($\mathcal{W}\triangle\mathcal{W}$-distance (Ben-David et al., 2010))**.** *For two distribution $\mathbb{P}_1$ and $\mathbb{P}_2$ over a domain $\mathcal{X}$ and a hypothesis class $\mathcal{W}$, the $\mathcal{W}\triangle\mathcal{W}$-distance between $\mathbb{P}_1$ and $\mathbb{P}_2$ w.r.t. $\mathcal{W}$ is defined as*

$$d_{\mathcal{W}\triangle\mathcal{W}}(\mathbb{P}_1, \mathbb{P}_2) = \sup_{\mathbf{w},\mathbf{w}'\in\mathcal{W}} \left|\mathbb{E}_{\mathbf{x}\sim\mathbb{P}_1}[f_{\mathbf{w}}(\mathbf{x}) \neq f_{\mathbf{w}'}(\mathbf{x})] - \mathbb{E}_{\mathbf{x}\sim\mathbb{P}_2}[f_{\mathbf{w}}(\mathbf{x}) \neq f_{\mathbf{w}'}(\mathbf{x})]\right| \tag{9}$$

**Definition 4** (Gradient-based Distribution Discrepancy)**.** *Given distributions $\mathbb{P}$ and $\mathbb{Q}$ defined over $\mathcal{X}$, the Gradient-based Distribution Discrepancy w.r.t. predictor $f_{\mathbf{w}}$ and loss $\ell$ is*

$$d_{\mathbf{w}}^{\ell}(\mathbb{P}, \mathbb{Q}) = \left\|\nabla R_{\mathbb{P}}(f_{\mathbf{w}}, \widehat{f}) - \nabla R_{\mathbb{Q}}(f_{\mathbf{w}}, \widehat{f})\right\|_2, \tag{10}$$

*where $\widehat{f}$ is a classifier which returns the closest one-hot vector of $f_{\mathbf{w}}$, $R_{\mathbb{P}}(f_{\mathbf{w}}, \widehat{f}) = \mathbb{E}_{\mathbf{x}\sim\mathbb{P}}\ell(f_{\mathbf{w}}, \widehat{f})$ and $R_{\mathbb{Q}}(f_{\mathbf{w}}, \widehat{f}) = \mathbb{E}_{\mathbf{x}\sim\mathbb{Q}}\ell(f_{\mathbf{w}}, \widehat{f})$.*

### I.3   Assumptions

**Assumption 1.**

- *The parameter space $\mathcal{W} \subset B(\mathbf{w}_0, r_1) \subset \mathbb{R}^d$ ($\ell_2$ ball of radius $r_1$ around $\mathbf{w}_0$);*

- *$\ell(f_{\mathbf{w}}(\mathbf{x}), y) \geq 0$ and $\ell(f_{\mathbf{w}}(\mathbf{x}), y)$ is $\beta_1$-smooth where $\ell(\cdot, \cdot)$ is the ID loss function;*

- *$\sup_{(\mathbf{x},y)\in\mathcal{X}\times\mathcal{Y}} \|\nabla\ell(f_{\mathbf{w}_0}(\mathbf{x}), y)\|_2 = b_1$, $\sup_{(\mathbf{x},y)\in\mathcal{X}\times\mathcal{Y}} \ell(f_{\mathbf{w}_0}(\mathbf{x}), y) = B_1$.*

*Remark* 1. For neural networks with smooth activation functions and softmax output function, we can check that the norm of the second derivative of the loss functions (cross-entropy loss and sigmoid loss) is bounded given the bounded parameter space, which implies that the $\beta$-smoothness of the loss functions can hold true. Therefore, our assumptions are reasonable in practice.

## J   Main Theorem

In this section, we provide a detailed and formal version of our main theorems with a complete description of the constant terms and other additional details that are omitted in the main paper.

**Theorem 2.** *Let $\mathcal{W}$ be a hypothesis space with a VC-dimension of $d$. Denote the datasets $\mathcal{S}^{in}$ and $\mathcal{S}^c_{selected}$ as the labeled in-distribution and the selected covariate OOD data by human feedback, and their sizes are $n$ and $m_c$, respectively. If $\widehat{\mathbf{w}} \in \mathcal{W}$ minimizes the empirical risk $R_{\mathcal{S}^c_{selected}}(f_{\mathbf{w}})$ of the multi-class classifier for classifying the covariate OOD, and $\mathbf{w}^* = \arg\min_{\mathbf{w} \in \mathcal{W}} R_{\mathbb{P}^{covariate}_{out}}(f_{\mathbf{w}})$, then for any $\delta \in (0,1)$, with probability of at least $1 - \delta$, we have*

$$R_{\mathbb{P}^{covariate}_{out}}(f_{\widehat{\mathbf{w}}}) \leq R_{\mathbb{P}^{covariate}_{out}}(f_{\mathbf{w}^*}) + 2\omega_{in}\sup_{\mathbf{w}\in\mathcal{W}} d^{\ell}_{\mathbf{w}}(\mathcal{S}^{in}, \mathcal{S}^c_{selected}) + 2\omega_{in}(2\sqrt{\frac{2d\log(2m_c) + \log\frac{2}{\delta}}{m_c}} + \gamma) + 2\zeta,$$

*where $\zeta = \sqrt{(\frac{\omega_{in}^2}{n} + \frac{\omega_c^2}{m_c})(\frac{d\log(2n+2m_c) - \log(\delta)}{2})} + \omega_{in}M$ and $\gamma = \min_{\mathbf{w}\in\mathcal{W}}\{R_{\mathbb{P}^{covariate}_{out}}(f_{\mathbf{w}}) + R_{\mathbb{P}_{in}}(f_{\mathbf{w}})\}$. $M$ is the upper bound of the loss function for the multi-class classifier*

$$\sup_{\mathbf{w}\in\mathcal{W}} \sup_{(\mathbf{x},y)\in\mathcal{X}\times\mathcal{Y}} \ell(f_{\mathbf{w}}(\mathbf{x}), y) \leq M. \tag{11}$$

*$\omega_{in}, \omega_c$ are two weight coefficients such that*

$$\underbrace{R_{\mathbb{P}_{in}, \mathbb{P}^{covariate}_{out}}(f_{\mathbf{w}})}_{Multi\text{-}class\ classifier} = \omega_{in}R_{\mathbb{P}_{in}}(f_{\mathbf{w}}) + \omega_c R_{\mathbb{P}^{covariate}_{out}}(f_{\mathbf{w}}), \tag{12}$$

*and $d^{\ell}_{\mathbf{w}}(\mathcal{S}^{in}, \mathcal{S}^c_{selected})$ is calculated as follows:*

$$d^{\ell}_{\mathbf{w}}(\mathcal{S}^{in}, \mathcal{S}^c_{selected}) = \|\nabla R_{\mathcal{S}^{in}}(f_{\mathbf{w}}, \widehat{f}) - \nabla R_{\mathcal{S}^c_{selected}}(f_{\mathbf{w}}, \widehat{f})\|_2,$$

*where $\widehat{f}$ is a classifier which returns the closest one-hot vector representation for the probabilistic prediction of $f_{\mathbf{w}}$, i.e., $R_{\mathcal{S}^{in}}(f_{\mathbf{w}}, \widehat{f}) = \mathbb{E}_{\mathbf{x}\sim\mathcal{S}^{in}}\ell(f_{\mathbf{w}}, \widehat{f})$ and $R_{\mathcal{S}^c_{selected}}(f_{\mathbf{w}}, \widehat{f}) = \mathbb{E}_{\mathbf{x}\sim\mathcal{S}^c_{selected}}\ell(f_{\mathbf{w}}, \widehat{f})$. Note that our theoretical analysis primarily focuses on the OOD generalization error of a specific set of covariate data, which is associated with the set $\mathcal{S}^c selected$. For simplicity, we will continue to use the notation $\mathbb{P}^{covariate}_{out}$.*

# K   Proof of the Main Theorem

In this section, we present the proof of our main Theorem 2. Before we dive into the proof details, we first clarify the analysis framework and the analysis target in our proof techniques.

Specifically, we consider the empirical error of the robust classification of samples from $\mathcal{S}^{\text{in}}$ and covariate OOD $\mathcal{S}^{\text{c}}_{\text{selected}}$ as the following weighted combination:

$$\underbrace{R_{\mathcal{S}^{\text{in}}, \mathcal{S}^{\text{c}}_{\text{selected}}}(f_{\mathbf{w}})}_{\text{Multi-class classifier}} = \omega_{\text{in}} R_{\mathcal{S}^{\text{in}}}(f_{\mathbf{w}}) + \omega_{\text{c}} R_{\mathcal{S}^{\text{c}}_{\text{selected}}}(f_{\mathbf{w}}). \tag{13}$$

Let $R_{\mathbb{P}_{\text{in}}}(f_{\mathbf{w}})$ represents the error of $f_{\mathbf{w}}$ on the in distribution (ID) data $\mathbb{P}_{\text{in}}$, and $R_{\mathbb{P}^{\text{covariate}}_{\text{out}}}(f_{\mathbf{w}})$ denotes the error of $f_{\mathbf{w}}$ on the covariate OOD data $\mathbb{P}^{\text{covariate}}_{\text{out}}$. $\omega_{\text{in}}$ and $\omega_{\text{c}}$ denote the weight coefficients. Similarly, we can define the true risk over the data distributions in the same way:

$$\underbrace{R_{\mathbb{P}_{\text{in}}, \mathbb{P}^{\text{covariate}}_{\text{out}}}(f_{\mathbf{w}})}_{\text{Multi-class classifier}} = \omega_{\text{in}} R_{\mathbb{P}_{\text{in}}}(f_{\mathbf{w}}) + \omega_{\text{c}} R_{\mathbb{P}^{\text{covariate}}_{\text{out}}}(f_{\mathbf{w}}). \tag{14}$$

**Step 1.** First, we prove that for any $\delta \in (0, 1)$ and $\mathbf{w} \in \mathcal{W}$, with probability of at least $1 - \delta$, we have

$$P[|R_{\mathbb{P}_{\text{in}}, \mathbb{P}^{\text{covariate}}_{\text{out}}}(f_{\mathbf{w}}) - R_{\mathcal{S}^{\text{in}}, \mathcal{S}^{\text{c}}_{\text{selected}}}(f_{\mathbf{w}})| \geq R] \leq \sqrt{(\frac{\omega_{\text{in}}^2}{n} + \frac{\omega_{\text{c}}^2}{m_{\text{c}}})(\frac{d \log (2n + 2m_{\text{c}}) - \log(\delta)}{2})}, \tag{15}$$

where $n, m_{\text{c}}$ are the sizes of datasets $\mathcal{S}^{\text{in}}, \mathcal{S}^{\text{c}}_{\text{selected}}$.

We first apply Theorem 3.2 of (Kifer et al., 2004) as restated in Lemma 8 to get the following equation,

$$P[|R_{\mathbb{P}_{\text{in}}, \mathbb{P}^{\text{covariate}}_{\text{out}}}(f_{\mathbf{w}}) - R_{\mathcal{S}^{\text{in}}, \mathcal{S}^{\text{c}}_{\text{selected}}}(f_{\mathbf{w}})| \geq R] \leq (2n + 2m_{\text{c}})^d \exp(\frac{-2R^2}{\frac{\omega_{\text{in}}^2}{n} + \frac{\omega_{\text{c}}^2}{m_{\text{c}}}}),$$

where $d$ is the VC dimension of the hypothesis space $\mathcal{W}$. Given $\delta \in (0, 1)$, we set the upper bound of the inequality to $\delta$, and solve for $R$:

$$\delta = (2n + 2m_{\text{c}})^d \exp(\frac{-2R^2}{\frac{\omega_{\text{in}}^2}{n} + \frac{\omega_{\text{c}}^2}{m_{\text{c}}}}).$$

We rewrite the inequality as

$$\frac{\delta}{(2n + 2m_{\text{c}})^d} = e^{-2R^2 / (\frac{\omega_{\text{in}}^2}{n} + \frac{\omega_{\text{c}}^2}{m_{\text{c}}})},$$

taking the logarithm of both sides, we get

$$\log \frac{\delta}{(2n + 2m_{\text{c}})^d} = -2R^2 / (\frac{\omega_{\text{in}}^2}{n} + \frac{\omega_{\text{c}}^2}{m_{\text{c}}}).$$

Rearranging the equation, we then get

$$R^2 = (\frac{\omega_{\text{in}}^2}{n} + \frac{\omega_{\text{c}}^2}{m_{\text{c}}})(\frac{d \log (2n + 2m_{\text{c}}) - \log(\delta)}{2}).$$

Therefore, with the probability of at least $1 - \delta$, we have

$$|R_{\mathbb{P}_{\text{in}}, \mathbb{P}^{\text{covariate}}_{\text{out}}}(f_{\mathbf{w}}) - R_{\mathcal{S}^{\text{in}}, \mathcal{S}^{\text{c}}_{\text{selected}}}(f_{\mathbf{w}})| \leq \sqrt{(\frac{\omega_{\text{in}}^2}{n} + \frac{\omega_{\text{c}}^2}{m_{\text{c}}})(\frac{d \log (2n + 2m_{\text{c}}) - \log(\delta)}{2})}. \tag{16}$$

**Step 2.** Based on Equation 16, we now prove Theorem 2. For the true error of hypothesis $\widehat{\mathbf{w}}$ on the covariate OOD data $R_{\mathbb{P}_{\text{out}}^{\text{covariate}}}(f_{\widehat{\mathbf{w}}})$, applying Lemma 7, Equation 16, and suppose $\mathbf{w}^* = \arg\min_{\mathbf{w} \in \mathcal{W}} R_{\mathbb{P}_{\text{out}}^{\text{covariate}}}(f_{\mathbf{w}})$, we get

$$
\begin{aligned}
R_{\mathbb{P}_{\text{out}}^{\text{covariate}}}(f_{\widehat{\mathbf{w}}}) &\leq R_{\mathbb{P}_{\text{in}}, \mathbb{P}_{\text{out}}^{\text{covariate}}}(f_{\widehat{\mathbf{w}}}) + \omega_{\text{in}}\big(\frac{1}{2}d_{\mathcal{W}\triangle\mathcal{W}}(\mathcal{S}^{\text{in}}, \mathcal{S}^{\text{c}}_{\text{selected}}) + 2\sqrt{\frac{2d\log(2m_{\text{c}}) + \log\frac{2}{\delta}}{m_{\text{c}}}} + \gamma\big) \\
&\leq R_{\mathcal{S}^{\text{in}}, \mathcal{S}^{\text{c}}_{\text{selected}}}(f_{\widehat{\mathbf{w}}}) + \sqrt{(\frac{\omega_{\text{in}}^2}{n} + \frac{\omega_{\text{c}}^2}{m_{\text{c}}})(\frac{d\log(2n + 2m_{\text{c}}) - \log(\delta)}{2})} \\
&\quad + \omega_{\text{in}}\big(\frac{1}{2}d_{\mathcal{W}\triangle\mathcal{W}}(\mathcal{S}^{\text{in}}, \mathcal{S}^{\text{c}}_{\text{selected}}) + 2\sqrt{\frac{2d\log(2m_{\text{c}}) + \log\frac{2}{\delta}}{m_{\text{c}}}} + \gamma\big) \\
&\leq R_{\mathcal{S}^{\text{in}}, \mathcal{S}^{\text{c}}_{\text{selected}}}(f_{\mathbf{w}^*}) + \sqrt{(\frac{\omega_{\text{in}}^2}{n} + \frac{\omega_{\text{c}}^2}{m_{\text{c}}})(\frac{d\log(2n + 2m_{\text{c}}) - \log(\delta)}{2})} \\
&\quad + \omega_{\text{in}}\big(\frac{1}{2}d_{\mathcal{W}\triangle\mathcal{W}}(\mathcal{S}^{\text{in}}, \mathcal{S}^{\text{c}}_{\text{selected}}) + 2\sqrt{\frac{2d\log(2m_{\text{c}}) + \log\frac{2}{\delta}}{m_{\text{c}}}} + \gamma\big) \\
&\leq R_{\mathbb{P}_{\text{in}}, \mathbb{P}_{\text{out}}^{\text{covariate}}}(f_{\mathbf{w}^*}) + 2\sqrt{(\frac{\omega_{\text{in}}^2}{n} + \frac{\omega_{\text{c}}^2}{m_{\text{c}}})(\frac{d\log(2n + 2m_{\text{c}}) - \log(\delta)}{2})} \\
&\quad + \omega_{\text{in}}\big(\frac{1}{2}d_{\mathcal{W}\triangle\mathcal{W}}(\mathcal{S}^{\text{in}}, \mathcal{S}^{\text{c}}_{\text{selected}}) + 2\sqrt{\frac{2d\log(2m_{\text{c}}) + \log\frac{2}{\delta}}{m_{\text{c}}}} + \gamma\big) \\
&\leq R_{\mathbb{P}_{\text{out}}^{\text{covariate}}}(f_{\mathbf{w}^*}) + 2\sqrt{(\frac{\omega_{\text{in}}^2}{n} + \frac{\omega_{\text{c}}^2}{m_{\text{c}}})(\frac{d\log(2n + 2m_{\text{c}}) - \log(\delta)}{2})} \\
&\quad + 2\omega_{\text{in}}\big(\frac{1}{2}d_{\mathcal{W}\triangle\mathcal{W}}(\mathcal{S}^{\text{in}}, \mathcal{S}^{\text{c}}_{\text{selected}}) + 2\sqrt{\frac{2d\log(2m_{\text{c}}) + \log\frac{2}{\delta}}{m_{\text{c}}}} + \gamma\big) \\
&= R_{\mathbb{P}_{\text{out}}^{\text{covariate}}}(f_{\mathbf{w}^*}) + 2\omega_{\text{in}}\big(\frac{1}{2}d_{\mathcal{W}\triangle\mathcal{W}}(\mathcal{S}^{\text{in}}, \mathcal{S}^{\text{c}}_{\text{selected}}) + 2\sqrt{\frac{2d\log(2m_{\text{c}}) + \log\frac{2}{\delta}}{m_{\text{c}}}} + \gamma\big) + 2\zeta_1,
\end{aligned}
$$

with probability of at least $1 - \delta$, where $\zeta_1 = \sqrt{(\frac{\omega_{\text{in}}^2}{n} + \frac{\omega_{\text{c}}^2}{m_{\text{c}}})(\frac{d\log(2n + 2m_{\text{c}}) - \log(\delta)}{2})}$ and $\gamma = \min_{h \in \mathcal{W}}\{R_{\mathbb{P}_{\text{out}}^{\text{covariate}}}(f_{\mathbf{w}}) + R_{\mathbb{P}_{\text{in}}}(f_{\mathbf{w}})\}$

**Step 3.** In this step, we aim to obtain the upper bound of the term $d_{\mathcal{W}\triangle\mathcal{W}}(\mathcal{S}^{\text{in}}, \mathcal{S}^{\text{c}}_{\text{selected}})$. To begin with, recall we have the following definition:

$$
d_{\mathcal{W}\triangle\mathcal{W}}(\mathcal{S}^{\text{in}}, \mathcal{S}^{\text{c}}_{\text{selected}}) = \sup_{\mathbf{w}, \mathbf{w}' \in \mathcal{W}} \big|\mathbb{E}_{\mathbf{x} \sim \mathcal{S}^{\text{in}}}[f_{\mathbf{w}}(\mathbf{x}) \neq f_{\mathbf{w}'}(\mathbf{x})] - \mathbb{E}_{\mathbf{x} \sim \mathcal{S}^{\text{c}}_{\text{selected}}}[f_{\mathbf{w}}(\mathbf{x}) \neq f_{\mathbf{w}'}(\mathbf{x})]\big|. \tag{17}
$$

Therefore, it is easy to check that

$$
\begin{aligned}
d_{\mathcal{W}\triangle\mathcal{W}}(\mathcal{S}^{\text{in}}, \mathcal{S}^{\text{c}}_{\text{selected}}) &= \sup_{\mathbf{w} \in \mathcal{W}} \big|R_{\mathcal{S}^{\text{in}}}(f_{\mathbf{w}}) - \nabla R_{\mathcal{S}^{\text{in}}}(f_{\mathbf{w}}, \widehat{f}) + \nabla R_{\mathcal{S}^{\text{in}}}(f_{\mathbf{w}}, \widehat{f}) - R_{\mathcal{S}^{\text{c}}_{\text{selected}}}(f_{\mathbf{w}}) \\
&\qquad - \nabla R_{\mathcal{S}^{\text{c}}_{\text{selected}}}(f_{\mathbf{w}}, \widehat{f}) + \nabla R_{\mathcal{S}^{\text{c}}_{\text{selected}}}(f_{\mathbf{w}}, \widehat{f})\big| \\
&\leq \sup_{\mathbf{w} \in \mathcal{W}} \big|R_{\mathcal{S}^{\text{in}}}(f_{\mathbf{w}}) - R_{\mathcal{S}^{\text{c}}_{\text{selected}}}(f_{\mathbf{w}})\big| + 2\sup_{\mathbf{w} \in \mathcal{W}} \big|\nabla R_{\mathcal{S}^{\text{in}}}(f_{\mathbf{w}}, \widehat{f}) - \nabla R_{\mathcal{S}^{\text{c}}_{\text{selected}}}(f_{\mathbf{w}}, \widehat{f})\big| \\
&\leq \sup_{\mathbf{w} \in \mathcal{W}} \big|R_{\mathcal{S}^{\text{in}}}(f_{\mathbf{w}})\big| + \sup_{\mathbf{w} \in \mathcal{W}} \big|R_{\mathcal{S}^{\text{c}}_{\text{selected}}}(f_{\mathbf{w}})\big| + 2\sup_{\mathbf{w} \in \mathcal{W}} d^{\ell}_{\mathbf{w}}(\mathcal{S}^{\text{in}}, \mathcal{S}^{\text{c}}_{\text{selected}}) \\
&\leq 2\sup_{\mathbf{w} \in \mathcal{W}} d^{\ell}_{\mathbf{w}}(\mathcal{S}^{\text{in}}, \mathcal{S}^{\text{c}}_{\text{selected}}) + 2M,
\end{aligned} \tag{18}
$$

where $\widehat{f}$ is a classifier which returns the closest one-hot vector of $f_{\mathbf{w}}$, $R_{\mathcal{S}^{\mathrm{in}}}(f_{\mathbf{w}}, \widehat{f}) = \mathbb{E}_{\mathbf{x} \sim \mathcal{S}^{\mathrm{in}}} \ell(f_{\mathbf{w}}, \widehat{f})$ and $R_{\mathcal{S}^{\mathrm{c}}_{\mathrm{selected}}}(f_{\mathbf{w}}, \widehat{f}) = \mathbb{E}_{\mathbf{x} \sim \mathcal{S}^{\mathrm{c}}_{\mathrm{selected}}} \ell(f_{\mathbf{w}}, \widehat{f})$.

The last inequality holds because of Proposition 3 and the definition of the Gradient-based Distribution Discrepancy in Definition 4. Therefore, we can prove that:

$$R_{\mathbb{P}^{\mathrm{covariate}}_{\mathrm{out}}}(f_{\widehat{\mathbf{w}}}) \leq R_{\mathbb{P}^{\mathrm{covariate}}_{\mathrm{out}}}(f_{\mathbf{w}^*}) + 2\omega_{\mathrm{in}} \sup_{\mathbf{w} \in \mathcal{W}} d^{\ell}_{\mathbf{w}}(\mathcal{S}^{\mathrm{in}}, \mathcal{S}^{\mathrm{c}}_{\mathrm{selected}}) + 2\omega_{\mathrm{in}}(4\sqrt{\frac{2d\log(2m_{\mathrm{c}}) + \log\frac{2}{\delta}}{m_{\mathrm{c}}}} + \gamma) + 2\zeta_1 + 2\omega_{\mathrm{in}}M. \tag{19}$$

# L    Necessary Lemmas, and Propositions

## L.1    Boundedness

*Proposition* 3. If Assumption 1 holds,

$$\sup_{\mathbf{w} \in \mathcal{W}} \sup_{(\mathbf{x},y) \in \mathcal{X} \times \mathcal{Y}} \|\nabla \ell(f_{\mathbf{w}}(\mathbf{x}), y)\|_2 \leq \beta_1 r_1 + b_1 = \sqrt{M'/2},$$

$$\sup_{\mathbf{w} \in \mathcal{W}} \sup_{(\mathbf{x},y) \in \mathcal{X} \times \mathcal{Y}} \ell(f_{\mathbf{w}}(\mathbf{x}), y) \leq \beta_1 r_1^2 + b_1 r_1 + B_1 = M,$$

*Proof.* One can prove this by *Mean Value Theorem of Integrals* easily. □

*Proposition* 4. If Assumption 1 holds, for any $\mathbf{w} \in \mathcal{W}$,

$$\left\| \nabla \ell(f_{\mathbf{w}}(\mathbf{x}), y) \right\|_2^2 \leq 2\beta_1 \ell(f_{\mathbf{w}}(\mathbf{x}), y).$$

*Proof.* The details of the self-bounding property can be found in Appendix B of (Lei & Ying, 2021). □

*Proposition* 5. If Assumption 1 holds, for any labeled data $\mathcal{S}$ and distribution $\mathbb{P}$,

$$\left\| \nabla R_{\mathcal{S}}(f_{\mathbf{w}}) \right\|_2^2 \leq 2\beta_1 R_{\mathcal{S}}(f_{\mathbf{w}}), \quad \forall \mathbf{w} \in \mathcal{W},$$

$$\left\| \nabla R_{\mathbb{P}}(f_{\mathbf{w}}) \right\|_2^2 \leq 2\beta_1 R_{\mathbb{P}}(f_{\mathbf{w}}), \quad \forall \mathbf{w} \in \mathcal{W}.$$

*Proof.* Jensen's inequality implies that $R_{\mathcal{S}}(f_{\mathbf{w}})$ and $R_{\mathbb{P}}(f_{\mathbf{w}})$ are $\beta_1$-smooth. Then Proposition 4 implies the results. □

## L.2    Necessary Lemmas for Theorem 2

*Lemma* 6 (Theorem 3.4 in Kifer et al. (2004)). Let $\mathcal{A}$ be a collection of subsets of some domain measure space, and assume that the VC-dimension is some finite $d$. Let $P_1$ and $P_2$ be probability distributions over that domain and $S_1$, $S_2$ finite samples of sizes $m_1$, $m_2$ drawn according to $P_1$, $P_2$ with certain selection criteria respectively. Then

$$P_{m1+m2}[|\phi_{\mathcal{A}}(S_1, S_2) - \phi_{\mathcal{A}}(P_1, P_2)| > R] \leq (2m_1)^d e^{-m_1 R^2/16} + (2m_2)^d e^{-m_2 R^2/16},$$

where $P_{m1+m2}$ is the $m1 + m2$'th power of $P$, the probability that $P$ induces over the choice of samples.

This theorem bounds the probability for the relativized discrepancy, and will help bounds the quantified distribution shifts between domains in our Theorem 2.

*Lemma* 7. Let $\mathcal{W}$ be a hypothesis space with a VC-dimension of $d$. Denote the datasets $\mathcal{S}^{\text{in}}$ and $\mathcal{S}^{\text{c}}_{\text{selected}}$ as the labeled in-distribution and the selected covariate OOD data, and their sizes are $n$ and $m_{\text{c}}$, respectively. Then for any $\delta \in (0, 1)$, for every $\mathbf{w} \in \mathcal{W}$ minimizing $R_{\mathcal{S}^{\text{in}}, \mathcal{S}^{\text{c}}_{\text{selected}}}(f_{\mathbf{w}})$ on datasets $\mathcal{S}^{\text{in}}, \mathcal{S}^{\text{c}}_{\text{selected}}$, we have

$$\|R_{\mathbb{P}_{\text{in}}, \mathbb{P}^{\text{covariate}}_{\text{out}}}(f_{\mathbf{w}}) - R_{\mathbb{P}^{\text{covariate}}_{\text{out}}}(f_{\mathbf{w}})\| \leq \omega_{\text{in}}(\frac{1}{2} d_{\mathcal{W} \triangle \mathcal{W}}(\mathcal{S}^{\text{in}}, \mathcal{S}^{\text{c}}_{\text{selected}}) + 4\sqrt{\frac{2d \log(2m_{\text{c}}) + \log \frac{2}{\delta}}{m_{\text{c}}}} + \gamma), \quad (20)$$

where $\gamma = \min_{\mathbf{w} \in \mathcal{W}} \{R_{\mathbb{P}_{\text{in}}}(f_{\mathbf{w}}) + R_{\mathbb{P}^{\text{covariate}}_{\text{out}}}(f_{\mathbf{w}})\}$. $d_{\mathcal{W} \triangle \mathcal{W}}(\mathcal{S}^{\text{in}}, \mathcal{S}^{\text{c}}_{\text{selected}})$ is defined according to Definition 3.

*Proof.* First, we prove that given datasets $\mathcal{S}^{\text{in}}, \mathcal{S}^{\text{c}}_{\text{selected}}$ from two distributions $\mathbb{P}_{\text{in}}$ and $\mathbb{P}^{\text{covariate}}_{\text{out}}$, we have

$$d_{\mathcal{W}\triangle\mathcal{W}}(\mathbb{P}_{\text{in}}, \mathbb{P}^{\text{covariate}}_{\text{out}}) \leq d_{\mathcal{W}\triangle\mathcal{W}}(\mathcal{S}^{\text{in}}, \mathcal{S}^{\text{c}}_{\text{selected}}) + 4\sqrt{\frac{2d\log(2m_{\text{c}}) + \log\frac{2}{\delta}}{m_{\text{c}}}}. \tag{21}$$

We start with Theorem 3.4 in Kifer et al. (2004), which is restated in Lemma 6:

$$P_{n+m_{\text{c}}}[|\phi_{\mathcal{A}}(\mathcal{S}^{\text{in}}, \mathcal{S}^{\text{c}}_{\text{selected}}) - \phi_{\mathcal{A}}(\mathbb{P}_{\text{in}}, \mathbb{P}^{\text{covariate}}_{\text{out}})| > R] \leq (2n)^d e^{-nR^2/16} + (2m_{\text{c}})^d e^{-m_{\text{c}}R^2/16}. \tag{22}$$

In this equation, $d$ is the VC-dimension of a collection of subsets of some domain measure space $\mathcal{A}$, while in our case, $d$ is the VC-dimension of hypothesis space $\mathcal{W}$. Following Ben-David et al. (2010), the VC-dimension of $\mathcal{W}\triangle\mathcal{W}$ is at most twice the VC-dimension of $\mathcal{W}$, and the VC-dimension of our domain measure space is thus $2d$.

Given $\delta \in (0, 1)$, we can set the upper bound of the inequality to $\delta$, and solve for $R$:

$$\delta = (2n)^{2d} \cdot e^{-nR^2/16} + (2m_{\text{c}})^{2d} \cdot e^{-m_{\text{c}}R^2/16}. \tag{23}$$

Let $n = m_{\text{c}}$, we can rewrite the inequality as:

$$\frac{\delta}{(2m_{\text{c}})^{2d}} = e^{-n\epsilon^2/16} + e^{-m_{\text{c}}\epsilon^2/16}, \tag{24}$$

taking the logarithm of both sides, we get

$$\log\frac{\delta}{(2m_{\text{c}})^{2d}} = -n\frac{\epsilon^2}{16} + \log\left(1 + e^{-(n-m_{\text{c}})\frac{\epsilon^2}{16}}\right), \tag{25}$$

rearranging the equation and defining $a = \frac{R^2}{16}$, we then get

$$\log\frac{\delta}{(2m_{\text{c}})^{2d}} = -m_{\text{c}}a + \log 2, \tag{26}$$

which implies

$$m_{\text{c}}a + \log(\delta/2) = 2d\log(2m_{\text{c}}). \tag{27}$$

Therefore, we have

$$R = 4\sqrt{a} = 4\sqrt{\frac{2d\log(2m_{\text{c}}) + \log\frac{2}{\delta}}{m_{\text{c}}}}. \tag{28}$$

With probability of at least $1 - \delta$, we have

$$|\phi_{\mathcal{A}}(\mathcal{S}^{\text{in}}, \mathcal{S}^{\text{c}}_{\text{selected}}) - \phi_{\mathcal{A}}(\mathbb{P}_{\text{in}}, \mathbb{P}^{\text{covariate}}_{\text{out}})| \leq 4\sqrt{\frac{2d\log(2m_{\text{c}}) + \log\frac{2}{\delta}}{m_{\text{c}}}}; \tag{29}$$

therefore,

$$d_{\mathcal{W}\triangle\mathcal{W}}(\mathbb{P}_{\text{in}}, \mathbb{P}^{\text{covariate}}_{\text{out}}) \leq d_{\mathcal{W}\triangle\mathcal{W}}(\mathcal{S}^{\text{in}}, \mathcal{S}^{\text{c}}_{\text{selected}}) + 4\sqrt{\frac{2d\log(2m_{\text{c}}) + \log\frac{2}{\delta}}{m_{\text{c}}}}. \tag{30}$$

Now in order to prove Lemma 7, we can use triangle inequality for classification error in the derivation.

For the true risk of hypothesis $f_{\mathbf{w}}$ on the covariate OOD data $R_{\mathbb{P}_{\text{out}}^{\text{covariate}}}(f_{\mathbf{w}})$, given the definition of $R_{\mathbb{P}_{\text{in}}, \mathbb{P}_{\text{out}}^{\text{covariate}}}(f_{\mathbf{w}})$,

$$
\begin{aligned}
\|R_{\mathbb{P}_{\text{in}}, \mathbb{P}_{\text{out}}^{\text{covariate}}}(f_{\mathbf{w}}) - R_{\mathbb{P}_{\text{out}}^{\text{covariate}}}(f_{\mathbf{w}})\| &= |\omega_{\text{in}} R_{\mathbb{P}_{\text{in}}}(f_{\mathbf{w}}) + \omega_{\text{c}} R_{\mathbb{P}_{\text{out}}^{\text{covariate}}}(f_{\mathbf{w}}) - R_{\mathbb{P}_{\text{out}}^{\text{covariate}}}(f_{\mathbf{w}})| \\
&\leq \omega_{\text{in}} |R_{\mathbb{P}_{\text{in}}}(f_{\mathbf{w}}) - R_{\mathbb{P}_{\text{out}}^{\text{covariate}}}(f_{\mathbf{w}})| \\
&\leq \omega_{\text{in}} (|R_{\mathbb{P}_{\text{in}}}(f_{\mathbf{w}}) - R_{\mathbb{P}_{\text{in}}}(f_{\mathbf{w}}, f_{\mathbf{w}^*})| + |R_{\mathbb{P}_{\text{in}}}(f_{\mathbf{w}}, f_{\mathbf{w}^*}) - R_{\mathbb{P}_{\text{out}}^{\text{covariate}}}(f_{\mathbf{w}}, f_{\mathbf{w}^*})| \\
&\quad + |R_{\mathbb{P}_{\text{out}}^{\text{covariate}}}(f_{\mathbf{w}}, f_{\mathbf{w}^*}) - R_{\mathbb{P}_{\text{out}}^{\text{covariate}}}(f_{\mathbf{w}})|) \\
&\leq \omega_{\text{in}} (R_{\mathbb{P}_{\text{in}}}(f_{\mathbf{w}^*}) + |R_{\mathbb{P}_{\text{in}}}(f_{\mathbf{w}}, f_{\mathbf{w}^*}) - R_{\mathbb{P}_{\text{out}}^{\text{covariate}}}(f_{\mathbf{w}}, f_{\mathbf{w}^*})| + R_{\mathbb{P}_{\text{out}}^{\text{covariate}}}(f_{\mathbf{w}^*})) \\
&\leq \omega_{\text{in}} (\gamma + |R_{\mathbb{P}_{\text{in}}}(f_{\mathbf{w}}, f_{\mathbf{w}^*}) - R_{\mathbb{P}_{\text{out}}^{\text{covariate}}}(f_{\mathbf{w}}, f_{\mathbf{w}^*})|),
\end{aligned}
$$

where $\gamma = \min_{h \in \mathcal{W}} \{R_{\mathbb{P}_{\text{in}}}(f_{\mathbf{w}}) + R_{\mathbb{P}_{\text{out}}^{\text{covariate}}}(f_{\mathbf{w}})\}$ and $f_{\mathbf{w}^*}$ is classifier that are parameterized with the optimal hypothesis $f_{\mathbf{w}^*}$ on $\mathbb{P}_{\text{in}}$. And we also have

$$
R_{\mathbb{P}_{\text{in}}}(f_{\mathbf{w}}, f_{\mathbf{w}^*}) = \mathbb{E}_{\mathbf{x} \sim \mathbb{P}}[|f_{\mathbf{w}}(\mathbf{x}) - f_{\mathbf{w}^*}(\mathbf{x})|]. \tag{31}
$$

By the definition of $\mathcal{W} \triangle \mathcal{W}$-distance and our proved Equation 30,

$$
\begin{aligned}
\|R_{\mathbb{P}_{\text{in}}}(f_{\mathbf{w}}, f_{\mathbf{w}^*}) - R_{\mathbb{P}_{\text{out}}^{\text{covariate}}}(f_{\mathbf{w}}, f_{\mathbf{w}^*})\| &\leq \sup_{\mathbf{w}, \mathbf{w}' \in \mathcal{W}} |R_{\mathbb{P}_{\text{in}}}(f_{\mathbf{w}}, f_{\mathbf{w}'}) - R_{\mathbb{P}_{\text{out}}^{\text{covariate}}}(f_{\mathbf{w}}, f_{\mathbf{w}'})| \\
&= \sup_{\mathbf{w}, \mathbf{w}' \in \mathcal{W}} P_{\mathbf{x} \sim \mathbb{P}_{\text{in}}}[f_{\mathbf{w}}(\mathbf{x}) \neq f_{\mathbf{w}^*}(\mathbf{x})] + P_{\mathbf{x} \sim \mathbb{P}_{\text{out}}^{\text{covariate}}}[f_{\mathbf{w}}(\mathbf{x}) \neq f_{\mathbf{w}^*}(\mathbf{x})] \\
&= \frac{1}{2} d_{\mathcal{W} \triangle \mathcal{W}}(\mathbb{P}_{\text{in}}, \mathbb{P}_{\text{out}}^{\text{covariate}}) \\
&\leq \frac{1}{2} d_{\mathcal{W} \triangle \mathcal{W}}(\mathcal{S}^{\text{in}}, \mathcal{S}_{\text{selected}}^{\text{c}}) + 2\sqrt{\frac{2d \log(2m_{\text{c}}) + \log \frac{2}{\delta}}{m_{\text{c}}}}.
\end{aligned}
$$

Therefore, we can get

$$
\begin{aligned}
|R_{\mathbb{P}_{\text{in}}, \mathbb{P}_{\text{out}}^{\text{covariate}}}(f_{\mathbf{w}}) - R_{\mathbb{P}_{\text{out}}^{\text{covariate}}}(f_{\mathbf{w}})| &\leq \omega_{\text{in}} (\gamma + |R_{\mathbb{P}_{\text{in}}}(f_{\mathbf{w}}, f_{\mathbf{w}^*}) - R_{\mathbb{P}_{\text{out}}^{\text{covariate}}}(f_{\mathbf{w}}, f_{\mathbf{w}^*})|) \\
&\leq \omega_{\text{in}} (\gamma + \frac{1}{2} d_{\mathcal{W} \triangle \mathcal{W}}(\mathbb{P}_{\text{in}}, \mathbb{P}_{\text{out}}^{\text{covariate}})) \\
&\leq \omega_{\text{in}} (\frac{1}{2} d_{\mathcal{W} \triangle \mathcal{W}}(\mathcal{S}^{\text{in}}, \mathcal{S}_{\text{selected}}^{\text{c}}) + 2\sqrt{\frac{2d \log(2m_{\text{c}}) + \log \frac{2}{\delta}}{m_{\text{c}}}} + \gamma),
\end{aligned}
$$

with probability of at least $1 - \delta$, where $\gamma = \min_{\mathbf{w} \in \mathcal{W}} \{R_{\mathbb{P}_{\text{in}}} + R_{\mathbb{P}_{\text{out}}^{\text{covariate}}}(f_{\mathbf{w}})\}$. This completes the proof.

$\square$

*Lemma* 8. Under the same conditions as Lemma 7, if the empirical risk is denoted as $R_{\mathcal{S}^{\text{in}}, \mathcal{S}_{\text{selected}}^{\text{c}}}(f_{\mathbf{w}})$ (as defined in Equation 13), then for any $\delta \in (0, 1)$ and $\mathbf{w} \in \mathcal{W}$, with the probability of at least $1 - \delta$, we have

$$
P[|R_{\mathbb{P}_{\text{in}}, \mathbb{P}_{\text{out}}^{\text{covariate}}}(f_{\mathbf{w}}) - R_{\mathcal{S}^{\text{in}}, \mathcal{S}_{\text{selected}}^{\text{c}}}(f_{\mathbf{w}})| \geq R] \leq 2 \exp(\frac{-2R^2}{\frac{\omega_{\text{in}}^2}{n} + \frac{\omega_{\text{c}}^2}{m_{\text{c}}}}) \tag{32}
$$

*Proof.* We apply Hoeffding's Inequality in our proof. Specifically, denote the true labeling function as $f_{\bar{\mathbf{w}}}$, we have that

$$\mathbb{P}(|[\sum_{i=0}^{n}\frac{\omega_{\mathrm{in}}}{n}|f_{\mathbf{w}}(\mathbf{x}_i)-f_{\bar{\mathbf{w}}}(\mathbf{x}_i)| + \sum_{j=0}^{m_{\mathrm{c}}}\frac{\omega_{\mathrm{c}}}{m_{\mathrm{c}}}|f_{\mathbf{w}}(\mathbf{x}_j)-f_{\bar{\mathbf{w}}}(\mathbf{x}_j)|]-[\omega_{\mathrm{in}}\mathbb{E}_{\mathbf{x}\sim\mathbb{P}_{\mathrm{in}}}|f_{\mathbf{w}}(\mathbf{x})-f_{\bar{\mathbf{w}}}(\mathbf{x})| + \omega_{\mathrm{c}}\mathbb{E}_{x\sim\mathbb{P}_{\mathrm{c}}}|f_{\mathbf{w}}(\mathbf{x})-f_{\bar{\mathbf{w}}}(\mathbf{x})|]| \geq R)$$

$$\leq 2\exp(-\frac{2R^2}{\sum_{i=1}^{n}(b_i-a_i)^2}),$$

(33)

where $\frac{\omega_{\mathrm{in}}}{n}|f_{\mathbf{w}}(\mathbf{x}_i)-f_{\bar{\mathbf{w}}}(\mathbf{x}_i)| \in [0,\frac{\omega_{\mathrm{in}}}{n}]$ and $\frac{\omega_{\mathrm{c}}}{m_{\mathrm{c}}}|f_{\mathbf{w}}(\mathbf{x}_j)-f_{\bar{\mathbf{w}}}(\mathbf{x}_j)| \in [0,\frac{\omega_{\mathrm{c}}}{m_{\mathrm{c}}}]$.

Considering the weighted empirical error, we get

$$R_{\mathcal{S}^{\mathrm{in}},\mathcal{S}^{\mathrm{c}}_{\mathrm{selected}}}(f_{\mathbf{w}}) = \omega_{\mathrm{in}}R_{\mathcal{S}^{\mathrm{in}}}(f_{\mathbf{w}}) + \omega_{\mathrm{c}}R_{\mathcal{S}^{\mathrm{c}}_{\mathrm{selected}}}(f_{\mathbf{w}})$$

$$= \sum_{i=0}^{n}\frac{\omega_{\mathrm{in}}}{n}|f_{\mathbf{w}}(\mathbf{x}_i)-f_{\bar{\mathbf{w}}}(\mathbf{x}_i)| + \sum_{j=0}^{m_{\mathrm{c}}}\frac{\omega_{\mathrm{c}}}{m_{\mathrm{c}}}|f_{\mathbf{w}}(\mathbf{x}_j)-f_{\bar{\mathbf{w}}}(\mathbf{x}_j)|,$$

which corresponds to the first part of Hoeffding's Inequality.

Due to the linearity of expectations, we can calculate the sum of expectations as

$$\omega_{\mathrm{in}}\mathbb{E}_{\mathbf{x}\sim\mathbb{P}_{\mathrm{in}}}|f_{\mathbf{w}}(\mathbf{x})-f_{\bar{\mathbf{w}}}(\mathbf{x})| + \omega_{\mathrm{c}}\mathbb{E}_{x\sim\mathbb{P}_{\mathrm{c}}}|f_{\mathbf{w}}(\mathbf{x})-f_{\bar{\mathbf{w}}}(\mathbf{x})| = \omega_{\mathrm{in}}R_{\mathbb{P}_{\mathrm{in}}}(f_{\mathbf{w}}) + \omega_{\mathrm{c}}R_{\mathbb{P}^{\mathrm{covariate}}_{\mathrm{out}}}(f_{\mathbf{w}}) = R_{\mathbb{P}_{\mathrm{in}},\mathbb{P}^{\mathrm{covariate}}_{\mathrm{out}}}(f_{\mathbf{w}}),$$

which corresponds to the second part of Hoeffding's Inequality. Therefore, we can apply Hoeffding's Inequality as

$$P[|R_{\mathbb{P}_{\mathrm{in}},\mathbb{P}^{\mathrm{covariate}}_{\mathrm{out}}}(f_{\mathbf{w}}) - R_{\mathcal{S}^{\mathrm{in}},\mathcal{S}^{\mathrm{c}}_{\mathrm{selected}}}(f_{\mathbf{w}})| \geq R] \leq 2\exp(\frac{-2R^2}{\frac{\omega_{\mathrm{in}}^2}{n}+\frac{\omega_{\mathrm{c}}^2}{m_{\mathrm{c}}}}).$$

$\square$

# M    Verification of Main Theorem

**Optimal Loss for Covariate OOD.**  For training the model, we utilized 50,000 covariate OOD data samples. The optimal loss for covariate OOD data, denoted as $R_{\mathbb{P}^{\mathrm{covariate}}_{\mathrm{out}}}(f_{\mathbf{w}})$, was evaluated using the CIFAR-10 versus CIFAR-10-C datasets (with Gaussian noise). The results indicated an optimal loss of 0.2383 on the test set, with a corresponding OOD test accuracy of 92.79%. This small optimal loss for covariate OOD data contributes to a tighter upper bound.

**Optimal Loss for In-Distribution (ID) Data.**  The training involved 50,000 ID data samples to determine the optimal loss for ID data, represented as $R_{\mathbb{P}_{\mathrm{in}}}(f_{\mathbf{w}})$. In the CIFAR-10 context, the optimal loss on the test set for ID data was recorded as 0.1792, while the corresponding ID accuracy on test data reached 95.13%. The minimal nature of the optimal loss for ID data is consistent with expectations and results in a tighter upper bound.

**Gradient Discrepancy.**  The gradient discrepancy for the ID CIFAR-10, Covariate OOD CIFAR-10-C (Gaussian noise), and Semantic OOD Textures dataset was found to be 0.00035. This small gradient discrepancy suggests a tighter upper bound.

| Dataset | Gradient Discrepancy↓ | OOD Acc.↑ |
|---|---|---|
| **CIFAR-10-C (Gaussian noise)** | 0.00035 | 90.37 |
| **CIFAR-10-C (Shot noise)** | 0.00030 | 82.04 |
| **CIFAR-10-C (Glass blur)** | 0.00040 | 92.41 |

Table 11: Empirical verification of gradient discrepancy in Theorem 1.

**Gradient Discrepancy Versus Covariate OOD Accuracy Across Different Datasets.** Table 11 offers a comparative analysis, empirically validating the gradient discrepancy among various datasets. The results show a correlation between gradient discrepancy and OOD accuracy.

## N   Impact Statements and Limitations

**Broader Impact**. Our research aims to raise both research and societal awareness regarding the critical challenges posed by OOD detection and generalization in real-world contexts. On a practical level, our study has the potential to yield direct benefits and societal impacts by ensuring the safety and robustness of deploying classification models in dynamic environments. This is particularly valuable in scenarios where practitioners have access to unlabeled datasets and need to discern the most relevant portions for safety-critical applications, such as autonomous driving and healthcare data analysis. From a theoretical standpoint, our analysis contributes to a deeper understanding of leveraging unlabeled wild data by using gradient-based scoring for selecting the most informative samples for human feedback. In Appendix M, we properly verify the necessary conditions of our bound using real-world datasets. Hence, we believe our theoretical framework has a broad utility and significance.

**Limitations**. Our proposed algorithm aims to improve both out-of-distribution detection and generalization results by leveraging unlabeled data. It still requires a small amount of human annotations and an additional gradient-based scoring procedure for deployment in the wild. Therefore, extending our framework to further reduce the annotation and training costs is a promising next step.

## O   Software and Hardware

We run all experiments with Python 3.8.5 and PyTorch 1.13.1, using NVIDIA GeForce RTX 2080 Ti GPUs.

