# OpenReview forum: "Out-of-Distribution Learning with Human Feedback"
_TMLR — Accepted by TMLR_

### Review · Reviewer_vUE7 · 2024-09-19

**Summary Of Contributions:**

This paper proposes a novel framework for classification tasks with out-of-distribution (OOD) samples (both semantic and covariate). The algorithm learns to cope with the OOD samples with relatively few human labels and a large unlabeled dataset. Superiority of the algorithm is demonstrated via sound empirical experiments and partial theoretical analysis.

**Audience:**

Yes

**Broader Impact Concerns:**

Nnone of those are considered as necessary to be specifically highlighted here.

**Claims And Evidence:**

Yes

**Requested Changes:**

The authors should address the above weaknesses, which are critical for acceptance.

**Strengths And Weaknesses:**

Strengths:
- The paper proposes an interesting idea to leverage large unlabeled datasets via only a few human labeled samples.
- The algorithm outperforms previous methods in the designed experiments.
- The flow of the paper is smooth, and the claims are clear.

Weaknesses:
- Technical details should be correct and accurate. For example, in Equation (4), the dimension of matrix G seems to be $\text{dim}(w)m\times 1$, which contradicts with the fact that its singular vector should has dimension $\text{dim}(w)$ (from Equation (5)). I believe the dimension of $G$ should be $\text{dim}\times m$, and its definition should be changed accordingly. Another example is from Theorem 1, where $\hat{f}$ is not clearly defined. Maybe I'm missing something here, but it's not clear to me now what "the closest one-hot vector of $f_w$" means.
- More explanation about the theoretical results should be provided. Currently, it seems the term $d_w^l$ doesn't decay as the sample size grows. If so, explanation on how large this term will be helpful, e.g. calculate the term using basis distributions including gaussians.

---

> ### Author Response · Authors · 2024-10-29
> **Response to Reviewer vUE7**
>
> Thank you for the constructive feedback. We have carefully considered each point and provide responses below in detail.
>
> > *W1-1. Clarification on the dimension of matrix $G$.*
>
> In Equation (4), the matrix $G$ has dimension $m \times \text{dim}(w)$, where $m$ is the number of samples in the wild dataset $S_{\text{wild}}$, and $\text{dim}(w)$ is the dimensionality of the gradient vector. This aligns with $G$ being a collection of gradient vectors for each wild data point, with each row representing the gradient difference for a specific sample.
>
> The top singular vector $v$ of $G$ is thus a vector of dimension $\text{dim}(w) \times 1$, as it lies in the gradient space and serves as the direction maximizing the projected gradient distances. We will make these dimensionalities explicit in the revised version to clarify this.
>
> > *W1-2. Definition of $\hat{f}$ and explanation of "the closest one-hot vector of $f_\omega$ in Theorem 1.*
>
> In Definition 4, the notation $\hat{f}$ represents a classifier that "returns the closest one-hot vector" of $f_w$. $\hat{f}$ projects $f_w$'s output, which is a vector of class probabilities, onto the cloest one-hot representation. Specifically,  the entry with the highest probability in the output of $f_w$ is set to 1, and all other entries are set to 0, aligning the prediction with a single class. We will clarify this in the revised manuscript to ensure clarity.
>
>  > *W2. More explanation about the theoretical results regarding $d_w^l$.*
>
> Thank you for highlighting the need for further clarification on the gradient discrepancy term $d_w^l$. As shown in Appendix L of our paper, we provided empirical validation of $d_w^l$ by measuring the gradient discrepancy across various datasets, including the ID (CIFAR-10), covariate OOD (CIFAR-10-C) with Gaussian noise, and semantic OOD (Textures) dataset. The observed gradient discrepancy values were notably small, such as 0.00035 for CIFAR-10-C (Gaussian noise) with a corresponding OOD accuracy of 90.37%. This suggests a tight upper bound in our theoretical results.
>
> The consistently small values across different shifts, including Gaussian noise, Shot noise, and Glass blur, as detailed in Appendix L, reinforce that $d_w^l$ remains bounded. This aligns well with our theoretical framework and enhances the robustness of our model under diverse distributional conditions.

---

### Review · Reviewer_FhLH · 2024-11-25

**Summary Of Contributions:**

The authors introduce a framework for OOD learning with human feedback.
* Following a previous work, they assume that we have access to labelled ID data and unlabelled wild data. The wild data is a mixture of ID, covariate OOD, and semantic OOD data.
* The proposed framework employs a gradient-based sample selection mechanism, which selects a small number of informative samples from the wild data distribution.
* The selected samples will be labelled by humans and then used to train a robust ID classifier and an OOD detector.

The authors also provide theoretical insights on the generalization error bounds to justify their algorithm. The proposed method outperforms previous ones in experiments.

**Audience:**

Yes

**Broader Impact Concerns:**

The paper tackles significant challenges in OOD detection and generalization with wild data, which are relevance to many applications. There are no concerns regarding ethical implications.

**Claims And Evidence:**

Yes

**Requested Changes:**

Please see the major weaknesses.

**Strengths And Weaknesses:**

## Strengths
* The paper is well-written and easy to understand.
* The experiments are comprehensive, including OOD detection, OOD generalization, and in-depth analysis.
 * The proposed algorithm shows significant improvement over previous methods.

## Weaknesses
### Major
* I am confused with the term “human feedback”. It feels like the authors are using some reinforcement learning techniques. For example, given an input, the model generates some output, receive human feedback, and improve itself.  I think this work is more aligned with active learning. It involves selecting a set of informative unlabelled samples, asking humans to label these samples, and retraining / fine-tuning the model. These fields may share similar ideas, so I hope the authors could clarify this.
* I am not familiar with OOD learning with in-the-wild data, such as Woods and Scone. I am wondering if these previous works use extra labels from humans in wild data to train their model, or if they just use the unlabelled wild data. If the authors collect extra labels from humans but they do not, are Tables 1 & 2 showing fair comparison?
* How do the authors decide $\pi_c$ and $\pi_s$, the mixture ratios of covariate and semantic OOD data in unlabelled wild data? Will these two ratios have large effect on method performance? I notice that the authors fixed them in the experiments, $\pi_c=0.5$ and $\pi_s=0.1$. Could the authors clarify this?

### Minor
* The citation styles need to be improved. For instance, in Page 9, “Impact of different sampling scores.”. The authors use textual citations (\citet) but these should be parenthetical citations (\citep). While in Related Works, some sentences like “The work of (Das et al., 2023; …” The authors use parenthetical citations but there should be textual citations.
* I’m curious about the effect of the top singular vector of gradient matrix. Is it not good if the authors simply calculate the distance between gradient of one sample and the average gradient, or use the average of more singular vectors?
* After receiving human feedback, do the algorithm use the remaining unlabelled wild data or throw them away? Is it better to utilize the remaining data with some loss functions?

---

> ### Author Response · Authors · 2024-12-15
> **Response to Reviewer FhLH**
>
> Thank you for your detailed feedback. We address your concerns below in details.
>
>  > *W1. Clarification of the term "human feedback".*
>
> The term "human feedback" encompasses several concepts studied in literature, including human-in-the-loop learning, interactive learning, bandits, and active learning. The term "human feedback" reflects our finding: OOD learning with human feedback can significantly improve OOD generalization and detection performance.
>
> We further discuss active learning in Appendix C. Unlike traditional active learning, which involves iterative querying and training, our approach requires only a single fine-tuning step. Moreover, existing active learning methods do not address OOD robustness in realistic wild data scenarios, which is a major focus of our framework.
>
>
>  > *W2. Comparison with OOD learning with in-the-wild data, such as Woods and Scone.*
>
> Recent works such as Woods and SCONE address similar learning with wild data settings, and thus we include these as baselines for comparison. Additionally, we compare various sampling scores and random sampling baselines with same amount of extra labels from humans in Table 4, demonstrating the effectiveness of our proposed method.
>
>  > *W3. How to decide the mixture ratios of covariate and semantic OOD data in the unlabeled wild data.*
>
> Our method does not assume specific mixture ratios of covariate and semantic OOD data within the wild dataset. The proposed approach performs robustly across different mixing ratios, indicating its broad applicability.
>
>
> > *W4. The citation styles need to be improved.*
>
> We will revise this accordingly.
>
> > *W5. Effect of the top singular vector of gradient matrix.*
>
> The top singular vector of the gradient matrix is a principled way to identify the strongest gradient deviation direction. By projecting each sample's gradient onto this vector, we capture the most significant deviation relative to the in-distribution gradients.
>
> > *W6. After receiving human feedback, do the algorithm use the remaining unlabeled wild data or throw them away?*
>
> Our approach can be extended to perform selection, labeling, and fine-tuning iteratively. However, we perform only a single iteration in this work since it already achieves strong performance. Extending to multiple iterations is a possible direction for future work.

---

### Review · Reviewer_Gme9 · 2024-12-01

**Summary Of Contributions:**

This paper proposes to tackle out-of-distribution (OOD) learning with human feed-back.  Under a limited annotation budget, the proposed method increasingly annotates unlabeled wild data, aiming to address the challenges of OOD generalization and OOD detection.  The core idea is to sample the most informative wild data in each annotation stage, given the current knowledge of labeled data.  The informativeness is measured as the norm of coefficients w.r.t the major principal components of the gradient matrix.  The experiment demonstrates superior OOD classification and OOD detection performance against other strong baselines.

**Audience:**

Yes

**Claims And Evidence:**

Yes

**Requested Changes:**

1. Please include discussion of active learning in the introduction/method section
2. Please add the reference according to Weakness 2.
3. Please address my concern of Weakness 3.
4. Please distinguish covariate and semantic OOD in the evaluation metric.
5. Please add discussion about OOD generalization and foundation models.

**Strengths And Weaknesses:**

Strengths:
1. The idea of measuring *informativeness* with the norm of coefficients w.r.t the major principal components of the gradient matrix is intriguing.  It follows the similar idea as PCA, where the major principal components maximally explains the data distribution, which is the loss gradients in this paper.  The loss is calculated with pseudo labels, and thus, is equivalent to "measurement of confidence".
2. The experiment results support the proposition of the paper, showing better OOD generalization and OOD detection performance.
3. The paper is well written and easy to follow

---

Weaknesses:

1. The problem formulation of incremental annotation is similar to the idea of active learning.  However, the paper does not discuss clearly in the introduction/method section.
2. The statement of "Previous works on OOD learning often rely heavily on statistical approaches or predefined assumptions
about OOD data distributions, which may not accurately reflect the complexity and diversity of real-world
scenarios." is not supported by any reference.
3. In Eqn 4, why does the gradient matrix $G$ of the wild data subtracts with the average gradient of the in-distribution data?
4. In table 1, the OOD Acc. metric does not distinguish semantic and covariate OOD. It's difficult to understand where does the performance gain come from.
5. The experiments are conducted only on Wide ResNet architecture pre-trained on Cifar-10.  However, there're already performance vision foundation models, e.g. CLIP.  Is the OOD generalization still a research problem in those fantastic foundation models, which are trained on extremely large-scale datasets?

---

> ### Author Response · Authors · 2024-12-15
> **Response to Reviewer Gme9**
>
> Thank you for your thoughtful feedback and valuable suggestions. We address your concerns below in detail.
>
>  > *W1. Include discussion of active learning.*
>
> We provide discussions on active learning in Appendix C. Classic active learning typically involves iterative training and querying process, while our proposed algorithm only requires a single model fine-tuning. Furthermore, existing deep active learning works do not study OOD robustness and the challenges posed by realistic scenarios involving wild data. Our proposed method is specifically tailored for both OOD generalization and detection challenges.
>
>  > *W2. Reference support for the statement: "Previous works do not accurately reflect the complexity and diversity of real-world scenarios.""*
>
> We provide references in Section 5 (Related Works), highlighting that prior research on OOD generalization primarily focuses on covariate shift, while OOD detection largely addresses semantic shift. Our work builds upon and extends these efforts by considering both covariate and semantic shifts jointly, which better reflects the complexities and diversity of real-world scenarios.
>
>  > *W3. In Equ 4, why does the gradient matrix G of the wild data subtracts with the average gradient of the in-distribution data?*
>
> We subtract the reference gradient to center the gradients of the wild dataset, which emphasizes deviations relative to the in-distribution data. This centering helps effectively identify and highlight OOD data.
>
>  > *W4. In table 1, clarification of OOD Acc.*
>
> The OOD Acc. metric refers to the classification accuracy on covariate OOD data. A higher OOD Acc. denotes better OOD generalization performance.
>
>  > *W5. Is OOD generalization still a research problem for foundation models trained on extremely large-scale datasets?*
>
> While advanced foundation models, such as CLIP, are trained on extremely large-scale datasets, they remain inherently biased toward the training data distribution. This bias does not guarantee the ability to recognize entirely unseen classes or handle OOD data effectively.
>
> For safety-critical applications, such as autonomous driving or medical imaging, OOD generalization remains crucial as even minor OOD failures can lead to severe consequences. Additionally, foundation models are often fine-tuned on task-specific datasets, which may introduce new domain shifts, making OOD detection and generalization critical.

---

### Decision · Action_Editor_jYeL · 2025-03-18

**Recommendation:** Accept with minor revision

**Comment:**

The paper received mixed reviews, with most reviewers acknowledging its strengths in tackling OOD learning through a structured human feedback mechanism.

The primary points of contention revolved around:
- The terminology of “human feedback” and its distinction from traditional active learning approaches.
- The need for clearer explanations regarding key theoretical claims, particularly in Appendix C.
- The evaluation methodology, including how OOD accuracy is measured and the applicability of the method to large-scale foundation models.

While one reviewer leaned toward rejection due to concerns over novelty and methodological clarity, others recommended acceptance with minor revisions. Given the overall positive recommendation and the authors’ willingness to address these concerns, a minor revision is recommended.

The authors should focus on:
- More explicitly distinguishing their method from active learning in the introduction and related work sections.
- Providing additional theoretical clarifications in Appendix C.
- Addressing specific concerns regarding metric definitions and evaluation fairness.
With these revisions, the paper would be a valuable contribution to the TMLR community.

**Audience:**

Yes, the topic of OOD learning is of significant interest to the TMLR community, especially given the increasing emphasis on robustness in machine learning models. The proposed integration of human feedback provides an interesting direction for improving OOD generalization, which is relevant to both theoretical and applied ML researchers.

**Claims And Evidence:**

The paper presents a novel framework for out-of-distribution (OOD) learning with human feedback, leveraging a selective annotation mechanism for improving OOD generalization and detection. The claims are generally well-supported with empirical evidence, including experiments demonstrating improvements over state-of-the-art baselines. However, there were requests for clarifications on theoretical claims, particularly regarding the generalization error bounds.

---

> ### Author Response · Authors · 2025-04-03
> **Final version submission**
>
> Dear Action Editor and Reviewers,
>
> Thank you for your time and effort in reviewing our submission and for your valuable feedback! We have now submitted the final version of our paper, incorporating the following revisions:
>
> - Added discussion on active learning and how our algorithm extends it (Last paragraph in the Introduction Section, revised Related Work Section)
> - Added references on previous works with strong and predefined statistical assumptions about OOD data distributions (Paragraph 2 in the Introduction Section)
> - Added clarification on metrics for different OOD data distributions (Caption of Table 1)
> - Added discussion on applicability on foundation models (Appendix Section H)
> - Added discussion on the dimension of matrix $\mathbf{G}$ (Footnote of Page 4)
> - Revised definition on the key concepts in Theorem 1 (the closest one-hot vector of $f_\mathbf{w}$)
> - Clarification / Verification on the gradient discrepancy between ID and unlabeled dataset in Theorem 1 (Appendix Section M)
> - Clarification on the fair comparison between the methods that have the same annotation budget (Table 4)
> - Revised Citation styles (Section 4.3)
>
> Best,
>
> Authors